# ADAPTIVE SPATIAL-TEMPORAL GENERALIZATION FOR PHYSICS-INFORMED NEURAL PDE SOLVERS

## ABSTRACT

Integrating domain knowledge into neural networks has advanced the development of Physics-Informed Neural Networks (PINNs), enabling solutions to partial differential equations across diverse applications. To enhance generalization in neural PDE solvers, we propose MagniLearning, an adaptive weighting strategy that dynamically adjusts the importance of spatial regions, knowledge components, and temporal segments during training. Our approach evaluates the impact of omitting each region or time block on model performance and assigns higher weights to the most influential data. This adaptive scheme accelerates convergence in neural PDE solvers by emphasizing the most informative regions and time segments, while enhancing robustness to noise and underrepresented physics. We formalize the method using an effective risk function that incorporates region- and time-dependent weights, and we provide theoretical guarantees for controlling the generalization error. Numerical experiments demonstrate that MagniLearning significantly improves both stability and accuracy.

## 1 INTRODUCTION

Neural networks represent a modern approach grounded in statistics and other mathematical concepts, profoundly impacting various fields of research and real-world applications. With the proliferation of neural networks in recent years, new strategies such as integrating domain knowledge into the learning process have been developed to improve their performance. Many of these innovations have proven particularly effective for solving Partial Differential Equations (PDEs), which are fundamental for modeling a wide range of physical phenomena across space and time. One such development, known as Physics-Informed Neural Networks (PINNs), introduced by Raissi et al. (2019), has demonstrated remarkable success in addressing a variety of PDE problems, making it a key tool in scientific machine learning. PINNs incorporate physics-based constraints directly into the loss function, effectively regularizing the neural network to avoid non-physical solutions. This method ensures that the network predictions remain consistent with the underlying physical principles, thus improving both accuracy and stability. Regularization techniques play a critical role in mitigating issues such as overfitting and numerical instability, which are particularly challenging in the context of solving PDEs.

PINNs excel in solving PDEs but face challenges with high-frequency, multi-scale, and large-scale problems. Domain decomposition addresses these issues by partitioning the computational domain into smaller subdomains solved independently while enforcing interface continuity. Inspired by Schwarz methods, multilevel domain decomposition enhances accuracy and efficiency, particularly for high-frequency and multi-scale scenarios Dolean et al. (2024). In fluid dynamics, it improves precision for incompressible Navier–Stokes equations by dynamically weighting subdomain solutions and mitigating gradient pathologies. Applications such as 2D Kovasznay flow and 3D blood flow simulations demonstrate significant gains in convergence and accuracy Gu et al. (2024); Shin et al. (2020); Gao et al. (2021); Du et al. (2019). For heterogeneous media, domain-adaptive PINNs (Da-PINNs) decompose domains along material interfaces, incorporating electromagnetic conditions into loss functions to optimize decomposition and reduce complexity Piao et al. (2024). Conservative PINNs (cPINNs) enforce flux continuity at subdomain interfaces, accelerating training and improving efficiency for conservation laws, as shown in Burgers' equation and lid-driven cavity problems using adaptive activation functions Jagtap et al. (2020). Domain decomposition also benefits inverse problems by localizing parameter estimation, enabling applications such as detecting nonuniform material damage Liu & Wu (2023).

Building on the theoretical foundation established by Yang & Ren (2022), who analyzed the convergence, generalization, and sampling complexity of integrating domain knowledge into neural networks, this study extends the framework in two key directions. First, we incorporate both time-dependent and time-independent partial differential equations (PDEs) as explicit domain knowledge within the informed loss formulation, enabling PINNs to more accurately capture temporal dynamics and steady-state behaviors in PDE-governed physical systems. Second, we propose a weighted loss function that adaptively adjusts the contributions of different informed loss components (e.g., labeled data, knowledge-based supervision), offering greater flexibility in balancing imperfect labels and imperfect knowledge. These enhancements not only broaden the applicability of informed learning for PDE-constrained problems but also improve robustness and tuning capability under noisy or incomplete supervision. Finally, we provide rigorous mathematical proofs and numerical experiments to demonstrate improved learning stability and accuracy of the proposed methods.

## 2 PRELIMINARIES AND PROBLEM FORMULATION

### 2.1 PRELIMINARIES

Consider the domain $D$ in dimensions $d$ (where $d = 1, 2$, or 3), with one dimension possibly representing time, making $D$ a spatial–temporal domain. We define a function $f(x, y)$ in this domain for points $(x, y) \in D$, which we aim to approximate using neural networks Dong & Li (2021); Ni & Dong (2023); Qian et al. (2024). We divide $D$ into $N_e$ (with $N_e \geq 1$) overlapping $\epsilon-$net sets $S_i$ as $D = S_1 \cup S_2 \cup \cdots \cup S_{N_e}$, where each $S_i$ represents the $i$-th $\epsilon-$net set. This overlapping formulation is motivated by the FBPINN Dolean et al. (2024), which combines local networks using partition of unity window functions to ensure smooth global solutions across subdomains.

Similarly to finite or spectral elements, we represent $f(x, y)$ locally on each $\epsilon-$net. Specifically, for each $\epsilon-$net set $S_i$ (for $1 \leq i \leq N_e$), $f(x, y)$ is represented by a feed-forward neural network. Let $f_i(x, y)$ (for $1 \leq i \leq N_e$) denote the function $f(x, y)$ restricted to the $\epsilon-$net set $S_i$. On any shared boundary $\Gamma_{ij}$ between $\epsilon-$net sets $S_i$ and $S_j$ (for all $1 \leq i, j \leq N_e$), we require that $f_i(x, y)$ and $f_j(x, y)$ satisfy $C^k$ continuity conditions with an appropriate multi-index $k = (k_1, k_2, \ldots, k_d)$. This means that their function values and partial derivatives up to order $k_s$ (for $1 \leq s \leq d$) should be continuous across the boundary and the outer regions of the $\epsilon-$net sets in the $s$-th direction. The order $k$ in the continuity $C^k$ is user-defined and, for differential equations, can be set according to the order of the equation in each coordinate direction.

For the feedforward neural network, let $h_i(x, y)$ $(1 \leq i \leq m)$ denote the output of the previous layer, and $h'_i(x, y) (1 \leq i \leq n)$ denote the output of this layer. Let $W$ and $L$ be integers, and suppose $1 \leq l_i \leq W$ $(0 \leq i \leq L)$ denotes $(L + 1)$ positive integers. Let $R \in \mathbb{R}$ represent a bounded positive real number, $z_0 \in \mathbb{R}^{l_0}$ denote the input variable, $\sigma : \mathbb{R} \to \mathbb{R}$ be a twice differentiable activation function, and $T_{W_k}$ $(1 \leq k \leq L)$ be an affine mapping $T_{W_k} : \mathbb{R}^{l_{k-1}} \to \mathbb{R}^{l_k}$ given by $z_{k-1} \mapsto T_{W_k}(z_{k-1}) = W_k z_{k-1} + b_k$ for $1 \leq k \leq L$, where $W_k \in [-R, R]^{l_k \times l_{k-1}} \subset \mathbb{R}^{l_k \times l_{k-1}}$ and $b_k \in [-R, R]^{l_k} \subset \mathbb{R}^{l_k}$ are the weight and bias parameters, respectively. Let $\Theta$ denote the collection of all parameters (weights and biases) defining these transformations, and $W \in \Theta$. The feedforward neural network is then defined as the mapping $h_{W^T, i} : \mathbb{R}^{l_0} \to \mathbb{R}^{l_L}$ given by $h_{W^T, i}(z_0) = T_{W_L} \circ \sigma \circ T_{W_{L-1}} \circ \cdots \circ \sigma \circ T_{W_1}(z_0)$, $z_0 \in \mathbb{R}^{l_0}$, where $\circ$ denotes the composition of the function. For the sake of clarity, we replace $h_{W^T, i}(z_0)$ with $h(x, y)$ Haghighat et al. (2020); Guo et al. (2020); Dashtbayaz et al. (2024).

### 2.2 PROBLEM DEFINITION AND PROPOSED FRAMEWORK

We consider a supervised learning scenario that involves two spaces, $\mathbf{X}$ and $\mathbf{Y}$, with the objective of learning a hypothesis function $h_W$ that predicts the output for the inputs $(x, y) \in S$. To train this function, we are given a dataset of $n$ samples, $\{(x_1, y_1), f(x_1, y_1), , \ldots, (x_n, y_n), f(x_n, y_n)\}$, where each $(x_i, y_i) \in S$ is an input and $f(x_i, y_i)$ is its corresponding target value, which the function $h(x_i, y_i)$ should predict. The training label $f(x_i, y_i)$ may not be the same as the true label $\hat{y}$ for the input $(x_i, y_i)$, because the training label may be of low quality (e.g., corrupted, noisy and / or quantized).

We assume a joint probability distribution $P((x, y), f(x, y))$ over spaces $\mathbf{X}$ and $\mathbf{Y}$, and that the training samples $n$ are independent and identically distributed (i.i.d.) from $P((x, y), f(x, y))$. This

probabilistic framework allows for the modeling of uncertainty in the predictions, since $f(x, y)$ is treated as a random variable conditioned on $(x, y)$, with distribution $P(f(x, y)|(x, y))$.

We define a real-valued nonnegative risk function $r(\hat{y}, z)$, as in Vapnik (1991), to quantify the deviation between the predicted value $z$ and the actual result $\hat{y}$. In classification tasks, such loss functions often act as scoring rules. The risk of a hypothesis $h(x, y)$ is the expected loss, expressed as $R(h) = \mathbb{E}[r(h(x, y), z)] = \int r(h(x, y), z)dP(x, y)$. The goal of a learning algorithm is to identify the hypothesis $h^*$ within a function class $\mathcal{H}$ that minimizes the risk: $h^* = \arg\min_{h \in \mathcal{H}} R(h)$.

We incorporate domain knowledge through a knowledge-based model $g(x, y)$ associated with input $(x, y)$ and define a knowledge-based risk function $r_K(h(x, y), g(x, y))$, which links the output of the deep neural network $h(x, y)$ to $g(x, y)$. The training risk of the informed neural network, called the informed risk, integrates this domain knowledge. For analytical tractability, we assume that both the risk function $r$ and the knowledge-based risk function $r_K$ are Lipschitz continuous, bounded above, and strongly convex with respect to the network output, consistent with prior assumptions in Xu et al. (2024); Zhou et al. (2023). Moreover, the eigenvalues of their Hessian matrices with respect to the network output are confined to the interval $[\rho, 1]$, where $\rho \in (0, 1]$.

To improve generalization in neural solvers for PDEs, the functions that to describe system dynamics in both spatial and temporal domains, and following prior work on adaptive weighting methods Gao et al. (2025); Michael et al. (2019), we introduce **MagniLearning**, a unified adaptive weighting framework that evaluates the importance of both spatial and temporal samples during training. Our method integrates three key components: (1) Leave-One-Region-Out (LORO) to compute region-specific weights, (2) Leave-One-Time-Out (LOTO) to evaluate the importance of different time blocks, and (3) Leave-One-Knowledge-Out (LOKO) to assess the contribution of unlabeled knowledge samples. These components estimate how removing a subset (region or time) affects the model's performance, allowing us to assign higher weights to more informative or challenging samples. The expected result is faster convergence, stronger generalization, and improved robustness.

We first define an effective optimal risk that combines region-specific weights from LORO ($\Gamma_i$) with time-based importance scores from LOTO ($\beta_t$). Specifically, for each smooth region and time block, we compute the weighted risk:

$$R_{\text{eff},k} = \alpha_t \sum_t \beta_t \gamma_i \sum_{i \in I_{\phi,k}} \Gamma_i \Big[ \mu_i r(h(x_i, y_i, t), f(x_i, y_i, t)) + (1 - \mu_i) r_k(h(x_i, y_i, t), g(x_i, y_i, t)) \Big]$$

(1)

where $\mu_i = \frac{1-\lambda}{n_z} \cdot \mathbf{1}(x_i \in S_z)$, $\lambda_i = \frac{\lambda}{n_g} \cdot \mathbf{1}(x_i \in S_g)$ and $\alpha_t$ and $\gamma_i$ are normalization terms to ensure balanced weighting. We extend the LORO+LOTO framework by introducing dynamic LOKO, which differentiates between supervised and unsupervised knowledge samples to compute adaptive region-level weights. We also use a dynamic coefficient $\kappa$ to adjust the influence of unsupervised samples based on their relative risk.

In the following Section 3, we present how this framework is applied to time-independent and time-dependent PDEs. Theorems 3.1 and 3.2 establish generalization guarantees for our proposed MagniLearning, providing theoretical support for the use of adaptive weighting to enhance learning stability and efficiency in neural PDE solvers.

## 3 METHODOLOGY

**Problem Setup** We begin by defining the governing partial differential equation (PDE), its domain, and boundary/initial conditions according to Dong & Li (2021); Raissi et al. (2019); Ni & Dong (2023). We then introduce a structured partitioning of the domain using an $\varepsilon$−net to guide sampling and local computations. Let $u$ denote the solution of the PDE, which is approximated by a neural network and referred to as the trial function. The operator $P$ denotes the differential operator acting on u, while $Q$ represents the source or forcing term. The function $g$ specifies the boundary or initial conditions, and the spatial domain is denoted by $D$, with boundary $\partial D$. For time-dependent problems, the full domain becomes $D \times (0, T]$, and the initial condition at $t = 0$ must be specified. The general form of a time-dependent PDE with a forcing term and boundary/initial conditions is:

$$\frac{\partial u}{\partial t} + Pu = Q, \quad \text{in } D \times (0, T]; \quad u = g, \quad \text{on } \partial D \times (0, T]; \quad u(x, 0) = u_0(x), \quad \text{in } D.$$

Note that the time-independent case can also be represented by the above equation, where the temporal derivative vanishes $\frac{\partial u}{\partial t} = 0$, reducing the PDE to a purely spatial formulation.

In the next step, representative points are selected across the domain using an $\varepsilon$-net, where each point $(x'_k, y'_k, t'_k)$ defines a localized region $S_k$ (denoted $C_{\phi,k}$) as a ball of radius $\phi$. The union of these regions approximately covers $D$, $D \approx \bigcup_{k=1}^{N_e} S_k$, with $N_e \sim \mathcal{O}(1/\phi^b)$, for some constant $b$. Collocation points are sampled within each $S_k$, and residuals are enforced locally inside and on its boundary. The $\varepsilon$-net construction from Yang & Ren (2022) replaces the non-overlapping fixed-grid partitioning in Dong & Li (2021) and remains well-defined even when regions overlap (Dolean et al., 2024). Using the separation method of Haeseler et al. (2017), the domain is fully covered by neighboring regions, each corresponding to an $\varepsilon$-net point and defined as a local ball of radius $\phi$.

**Definition 3.0. Set Construction:** Given $\phi > 0$, construct a $\varepsilon-$net set, where each separate circle has a radius of $\phi$. Define $U_\phi = \{(x'_k, y'_k) \mid k \in [N], (x'_k, y'_k) \in U\}$, where $N \sim O(1/\phi^b)$, so that for all $(x'_i, y'_i), (x'_j, y'_j) \in U_\phi$ and $(x'_i, y'_i) \neq (x'_j, y'_j)$, the following condition is satisfied: $\sqrt{(x'_i - x'_j)^2 + (y'_i - y'_j)^2} \geq \phi$. Furthermore, for any $(x_i, y_i) \in S_z \cup S_g$, where $S_z$ and $S_g$ are local neighbors as explained above, there exists at least one $(x'_k, y'_k) \in U_\phi$ satisfying $\sqrt{(x_i - x'_k)^2 + (y_i - y'_k)^2} \leq \phi$. Each input $(x', y') \in U_\phi$ is called a representative input and determines a smooth set $C_{\phi,k} = \{x \in X \mid \sqrt{(x - x'_j)^2 + (y - y'_j)^2} \leq \phi$ and $\phi/2 \leq \sqrt{(x - x'_k)^2 + (y - y'_k)^2}$ for all $j \neq k$, where $(x'_k, y'_k), (x'_j, y'_j) \in U_\phi\}$. The index set of training samples within the $k$-th smooth set is defined as $I_{\phi,k} = \{i \mid (x_i, y_i) \in S_z \cup S_g$ and $(x_i, y_i) \in C_{\phi,k}\}$.

We adopt the data separability assumption, which clusters training samples into smooth sets with similar input–output behavior. This milder alternative to point-wise separability allows samples to be arbitrarily close or even identical in input space. For each region, effective labels are defined to capture the combined influence of supervision and domain knowledge on network predictions. For the $k$-th smooth set, define the effective label as $y_{\text{eff},k} = \arg\min_h \sum_{i \in I_{\phi,k}} \left[ \mu_i r \left( h(x_i, y_i), f(x_i, y_i) \right) + \lambda_i r_k \left( h(x_i, y_i), g(x_i, y_i) \right) \right]$, with $\mu_i, \lambda_i$ defined and $h$ in the space of network output. The effective optimal risk is defined as $r_{\text{eff},k} = \sum_{i \in I_{\phi,k}} \left[ \mu_i r \left( y_{\text{eff},k}, f(x_i, y_i) \right) + \lambda_i r_k \left( y_{\text{eff},k}, g(x_i, y_i) \right) \right]$.

## 3.1 Learning from Leave-One-Region-Out

In standard neural network training, the objective is to minimize the loss over the training data, typically assuming that all regions $C_{\theta,k}$ contribute equally. We propose a learning method based on Leave-One-Region-Out (LORO) to reweight each region according to its influence on generalization. Specifically, LORO measures how informative or challenging each region is by evaluating the model's performance when that region is excluded during training. To implement this, we define a smooth partition of the dataset regions $C_{\phi,k} \subset D$, and construct a reduced dataset $D_{-k} = D \setminus C_{\phi,k}$ by removing the $k$-th region. The model is then trained on $D_{-k}$, yielding parameters $W_{-k}$. Using this model, we compute the effective risk $r_{\text{eff},k}^{\text{LORO}}$ for the excluded region by evaluating an effective label over the index set $I_{\phi,k}$ associated with region $C_{\phi,k}$: $y_{\text{eff},k}^{\text{LORO}}(\{W_{-k}\}_{i=1}^n) = \arg\min_h \sum_{i \in I_{\phi,k}} \left[ \mu_i r \left( h(x_i, y_i), f(x_i, y_i; W_{-k}) \right) + \lambda_i r_k \left( h(x_i, y_i), g(x_i, y_i; W_{-k}) \right) \right]$. Subsequently, through learning from LORO, the effective risk for region $\mathcal{C}_{\phi,k}$ is computed as: $r_{\text{eff},k}^{\text{LORO}}(\{W_{-k}\}_{i=1}^n) = \sum_{i \in I_{\phi,k}} \left[ \mu_i r \left( y_{\text{eff},k}^{\text{LORO}}, f(x_i, y_i; W_{-k}) \right) + \lambda_i r_k \left( y_{\text{eff},k}^{\text{LORO}}, g(x_i, y_i; W_{-k}) \right) \right]$. Note that in some numerical cases, we consider a sub-union of multiple regions $C_{\phi,k}$ instead of a single region $C_{\phi,k}$.

**LORO risk estimation:** The motivation of our approach is to adaptively reweight regions using the LORO mechanism. At each training iteration, the weight adjustment factor for region $C_{\phi,k}$ is computed as:

$$\alpha_i = \left[ \exp\left( -\beta \frac{r^{\text{eff},k} - r_{\text{eff},k}^{\text{LORO}}}{|r^{\text{eff},k}| + \epsilon} \right) \right]^\lambda,$$

where $r^{\text{eff},k}$ denotes the risk when region $C_{\phi,k}$ is included during training, and $r_{\text{eff},k}^{\text{LORO}}$ denotes the risk when it is excluded. The scaling constant $\beta$, $\lambda$ and the small positive constant $\epsilon$ ensure numerical

stability and regulate the sensitivity of the weight update. This formulation prioritizes regions that substantially reduce generalization risk, guiding the model to focus on more informative regions during optimization. See Appendix A for the full derivation and theoretical discussion.

**Definition 3.1** For the $k$-th smooth set, define the effective label as $y_{\text{eff},k} = \arg\min \gamma \sum_{i\in I_{\phi,k}} \Gamma_i \Big\{ \mu_i r\left(h(x_i,y_i), f(x_i,y_i)\right) + \lambda_i r_k\left(h(x_i,y_i), g(x_i,y_i)\right) \Big\}$, with $\mu_i$, $\lambda_i$ defined and $h$ in the space of network output. The effective optimal risk is defined as $\widehat{R}_{\text{LOO}}^{\Gamma} = \gamma \sum_{i\in I_{\phi,k}} \Gamma_i \Big\{ \mu_i\, r\left(y_{\text{eff},k}, f(x_i,y_i)\right) + \lambda_i\, r_k\left(y_{\text{eff},k}, g(x_i,y_i)\right) \Big\}$, where $\gamma = \frac{1}{\sum_i \Gamma_i}$.

Following the recommendations by Li et al. (2025), we present the following definitions.

**Definition 3.2** Define $n_L(n)$ and $n_K(n)$ as the power functions that control the influence of label-based and knowledge-based supervision at training stage $n$, respectively. We define the optimal hypothesis $h^*$ as:

$$h^* = \arg\min_h \gamma \sum_{n\in\mathbb{N}} \sum_{i\in\mathcal{I}_{\phi,L}} \left[ \Gamma_i^{\,n_L(n)} \mu_i r(h(x_i,y_i), f(x_i,y_i)) + \Gamma_i^{\,n_K(n)} \lambda_i r_K(h(x_i,y_i), g(x_i,y_i)) \right]$$

where the power schedules are defined such that the label weight decays with $n_L = N-n$ while the knowledge weight increases with $n_K = n$, gradually shifting the model's reliance from label-based to knowledge-based supervision over $n = 0, 1, \ldots, N$.

**Definition 3.3** Let the optimal hypothesis on the dataset $S$ be defined as: $h^* = \arg\min_h \sum_{S_g \cup S_z} \Gamma_i \left[ \mu_i\, r(h(x_i,y_i), f(x_i,y_i)) + \lambda_i\, r_K(h(x_i,y_i), g(x_i,y_i)) \right]$. We define the standard population risk on the reference dataset $S$, which contains true labels $Q_S^{\Gamma\text{-K-Eval}} = \sum_{S_g \cup S_z} r\left(h^*, \hat{y}_i\right)$.

**Theorem 3.1** *Assume $\phi \leq \widetilde{O}(\epsilon^2 L^{-9/2} \log^{-3}(m))$ and $\Phi \leq (\epsilon/n)^{1/b}$ and let complexity requirement is given by:*

$$m \geq \Omega\left(\phi^{-1} b^{-4} L^{15} d\rho^{-4} \bar{\lambda}^{-4} \alpha^{-4} \log^3(m)\right), \tag{2}$$

*where $\bar{\lambda}$ is defined as: $\bar{\lambda} = \Omega\left(\min(1-\lambda, \lambda) \cdot \mathbf{1}_{\lambda\in(0,1)} + \mathbf{1}_{\lambda\in\{0,1\}}\right)$. The step size $\eta$ for gradient descent is set as: $\eta = \mathcal{O}\left(\frac{d}{L^2 m}\right)$, after $T$ iterations, where $T = \mathcal{O}\left(\frac{L^2}{\phi^{1+2b}\rho\lambda\alpha} \log\left(\frac{1}{\epsilon} \log\left(\frac{1}{\phi}\right)\right)\right)$. Then, with probability at least $1 - O(\phi) - \delta$, for $\delta \in (0,1)$, the population risk satisfies the following:*

$$R(W^{(T)}) \leq O(\sqrt{\varepsilon})$$

$$+ Q_S^{\Gamma\text{-K-Eval}} + \mathcal{O}\left(\frac{L^{5/4}\phi^{1/2}\log^{1/4}(m)}{\sqrt{n}}\right) + O\left(\frac{\Phi'}{\sqrt{n}} + 3\sqrt{\frac{\log\frac{2}{\delta}}{2n}},\right)$$

*where $\Phi' = \sqrt{\gamma^2 \left(2H^{[\frac{1}{p^*} - \frac{1}{q}]+}\right)^{2(d-1)} \min\{p^*, 4\log(2D)\} \max_i \|x_i\|_{p^*}^2}$.*

### 3.2 LEARNING FROM LEAVE-ONE-KNOWLEDGE-OUT

In this section, we split the knowledge dataset into two subsets: supervised and unsupervised knowledge. The first subset, $S_g'$, contains samples that lie within the same smooth regions as labeled samples from $S_z$, while the second subset, $S_g''$, contains samples outside these regions. This separation clarifies each subset's contribution to the training objective. Building on the structured region-based framework of Yang et al. (2023), we extend their convergence- and risk-focused analysis by introducing adaptive sample weighting. In particular, we propose a Leave-One-Knowledge-Out (LOKO) validation strategy to assign weights according to informativeness and difficulty from knowledge information.

The first step in the LOKO procedure is to define the combined risk as: $R_{I,G} = (1-\beta)\frac{1}{n_z}\sum_{S_z} r(h(x_i,y_i), f(x_i,y_i)) + \beta\frac{1}{n_g'}\sum_{S_g'} r_K(h(x_i,y_i), g(x_i,y_i))$, and $R_{S'',I,G}(W) = $

$\frac{1}{n_g''} \sum_{S_g''} r_K(h(x_i, y_i), g(x_i, y_i))$ such that the total knowledge-guided risk satisfies: $R_{I,G}(W) = R_{S,I,G}(W) + R_{S'',I,G}(W)$.

**LOKO risk estimation:** Similar to LORO, the motivation for LOKO is to adaptively reweight unlabeled regions $S''$, which are the regions that are not supported by nearby labeled data. To implement this, we compute a regularized empirical loss over each excluded $k$-th unsupervised region using available supervision and the remaining region $S''_{-k}$. At each training iteration, we define the LOKO risk as: $\widehat{R}_{I,G}^{\text{LOKO}}(W) = (1 - \beta)\frac{1}{n_z} \sum_{z \in S_z} r(h(x_i, y_i), f(x_i, y_i)) + \beta \frac{1}{n_g} \sum_{g \in S_g'} r_K(h(x_i, y_i), g(x_i, y_i)) + R_{S'',I,G}^{\text{LOKO}}$, or $\widehat{R}_{I,G}^{\text{LOKO}}(W) = R_{S,I,G}(W) + R_{S'',I,G}^{\text{LOKO}}(W)$.

For each training iteration, we compute the weight adjustment factor for each unsupervised region:

$$\kappa_i = \begin{cases} \left( \dfrac{1}{1 + \exp\left( \alpha \left( R_{I,G}(W) - R_{I,G}^{\text{LOKO}}(W) \right) \right)} \right)^{\gamma}, & \text{if } \tau_{\min} \leq R_{I,G}^{\text{LOKO}}(W) \leq \tau_{\max} \\ 0, & \text{otherwise.} \end{cases}$$

where $\alpha$ is a sensitivity parameter controlling the steepness, $\gamma$ controls the scaling, and $\tau$ is the threshold beyond which the unsupervised component is disregarded. The average value is given by $\kappa = \frac{1}{n} \sum_{i=1}^{n} \kappa_i$. This formulation ensures:

- If $R_{I,G}(W) < R_{I,G}^{LOKO}$, then $\kappa \uparrow$: Emphasis on the unsupervised component.

- If $R_{I,G}(W) > R_{I,G}^{LOKO}$, then $\kappa \downarrow$: Emphasis on the supervised component.

The informed risk function incorporating the dynamic is defined as $\widehat{R}_{I,G}^{\kappa}(W) = \arg\min(1 - \kappa)(1 - \beta)\frac{1}{n_z} \sum_{S_z} r(h(x_i, y_i), f(x_i, y_i)) + (1 - \kappa)\beta \frac{1}{n_g'} \sum_{S_g'} r_K(h(x_i, y_i), g(x_i, y_i)) + \kappa \frac{1}{n_g''} \sum_{S_g''} r_K(h(x_i, y)), g(x_i, y_i))$. The parameter $\beta \in [0, 1]$ controls the relative importance of different sources of supervision. This formulation allows the model to dynamically adjust its learning focus based on the reliability of the supervision, placing greater emphasis on supervised or confident knowledge-based samples when $\kappa$ is small and gradually incorporating less confident supervision as $\kappa$ increases.

The parameter $\beta \in [0, 1]$ controls the relative importance of different supervision sources. This design enables the model to dynamically adjust its learning focus: when $\kappa$ is small, it prioritizes supervised or highly confident knowledge-based samples, and as $\kappa$ increases, it gradually incorporates less confident supervision.

**Theorem 3.2** *Assume $\phi \leq \widetilde{O}(\epsilon^2 L^{-9/2} \log^{-3}(m))$ and $\Phi \leq (\epsilon/n)^{1/b}$ and let complexity requirement is given by:*

$$m \geq \Omega\left( \phi^{-1} b^{-4} L^{15} d\rho^{-4} \bar{\lambda}^{-4} \alpha^{-4} \log^3(m) \right), \tag{3}$$

*where $\bar{\lambda}$ is defined as: $\bar{\lambda} = \Omega\left( \min(1 - \lambda, \lambda) \cdot \mathbf{1}_{\lambda \in (0,1)} + \mathbf{1}_{\lambda \in \{0,1\}} \right)$. The step size $\eta$ for gradient descent is set as: $\eta = \mathcal{O}\left( \frac{d}{L^2 m} \right)$, after $T$ iterations, where $T = \mathcal{O}\left( \frac{L^2}{\phi^{1+2b} \rho \bar{\lambda} \alpha} \log\left( \frac{1}{\epsilon} \log\left( \frac{1}{\phi} \right) \right) \right)$. Then, with probability at least $1 - O(\phi) - \delta$, for $\delta \in (0, 1)$, the population risk satisfies the following:*

$$R(W^T) \leq O(\sqrt{\epsilon}) + (1 - \kappa)\widehat{R}_S^{\kappa} + \kappa \widehat{R}_{S''}^{\kappa} + \mathcal{O}\left( \frac{L^{5/4} \phi^{1/2} \log^{1/4}(m)}{\sqrt{n}} \right)$$

$$+ O\left( \frac{\Phi'}{\sqrt{n}} + 3\sqrt{\frac{\log \frac{2}{\delta}}{2n}} \right) \tag{4}$$

*where $\Phi' = \sqrt{\gamma^2 \left( 2H^{[\frac{1}{p^*} - \frac{1}{q}]+} \right)^{2(d-1)} \min\{p^*, 4\log(2D)\} \max_i \|x_i\|_{p^*}^2}$, and the empirical risk is given by $\widehat{R}_S^{\kappa} = \frac{1}{n_z} \sum_{z \in S_z} r(h^*, \hat{y})$, where the optimal hypothesis is obtained by $h^* = \arg\min((1 - \beta)\frac{1}{n_z} \sum_{S_z} r(h(x_i, y_i), g(x_i, y_i)) + \beta \frac{1}{n_g'} \sum_{S_g'} r_K(h(x_i, y_i), f(x_i, y_i)))$. Similarly, the risk for the sample $S_g''$ is defined as $\widehat{R}_{S''}^{\kappa} = \frac{1}{n_z} \sum_{z \in S_z''} r(h^*, \hat{y})$ with the optimal hypothesis computed as $h^* = \frac{1}{n_g''} \sum_{x_i \in S_g''} r_K(h(x_i, y_i), g(x_i, y_i))$.*

## 3.3 LEARNING FROM LEAVE-ONE-TIME-OUT

In this section, we present an adaptive time-block weighting strategy based on estimated informativeness to improve learning efficiency in time-dependent neural PDE solvers. Rather than treating all blocks equally, we assign a learned weight to each block via a Leave-One-Time-Out (LOTO) risk evaluation, giving higher weights to those that contribute more to predictive accuracy. This focuses the model on the most informative time segments, enhancing stability and generalization over long horizons. Our approach is inspired by the block time marching (BTM) method Dong & Li (2021); Qian et al. (2024), which addresses the challenge of long-time dynamic simulation in time-dependent PDEs. Instead of solving over the entire time domain at once, we partition the temporal dimension into smaller intervals, or time blocks. Each block is solved independently, with the final state of one block serving as the initial condition for the next. This block-by-block progression stabilizes training and improves long-horizon accuracy.

Here to define the LOTO mechanism, we first formulate a time-dependent risk function. Let $I_{\phi,k}$ be the index set of samples within the $k$-th smooth set $C_{\phi,k}$, and let $T_i$ denote the temporal extent for each block $i$. The optimal risk function $y_{\text{eff},k}$ in the continuous case is then defined as follows.

**Definition 3.4** The effective label calculation for learning from LOTO is given by: $y_{\text{eff},k} = \arg\min \sum_{t=1}^{T_i} \int_S [\mu(x,y) r(h(x,y,t) - f(x,y,t)) + (1 - \mu(x,y)) r_k(h(x,y,t) - g(x,y,t))] \, dS \, dt$. The effective optimal risk function is defined as: $r_{\text{eff},k}^{full} = \sum_{t=1}^{T_i} \int_S [\mu(x,y) r(y_{\text{eff},k}, f(x,y,t)) + (1 - \mu(x,y)) r_k(y_{\text{eff},k}, g(x,y,t))] \, dS \, dt$.

**LOTO risk estimation:** Assume the temporal dimension is partitioned into sequential time blocks, denoted as *Block 1, Block 2, ..., Block $N+1$*, where each block corresponds to a specific segment of time. We first compute the risk using all $N+1$ blocks, then remove *Block $N+1$* and recompute the risk using only the first $N$ blocks. The new effective label for LOTO is then calculated as: $y_{\text{eff},k}^{\text{LOTO-N+1}} = \arg\min \sum_{t=t_1}^{t_{N+1}} \int_S \left[ \mu(x,y) \, r \left( h(x,y,t_{\text{LOTO-Time}_{N+1}}), f(x,y,t_{\text{LOTO-Time}_{N+1}}) \right) + (1 - \mu(x,y)) \, r_K \left( f(x,y,t_{\text{LOTO-Time}_{N+1}}), g(x,y,t_{\text{LOTO-Time}_{N+1}}) \right) \right] dS$. The LOTO effective optimal risk is defined as: $R_{\text{eff},(N+1)}^{\text{LOTO-N+1}} = \sum_{t=t_1}^{t_2} \int_S \left[ \mu(x,y) \, r \left( y_{\text{eff},k}^{\text{LOTO-N+1}}, f(x,y,t_{\text{LOTO-Time-(N+1)}}) \right) + (1 - \mu(x,y)) \, r_K \left( y_{\text{eff},k}^{\text{LOTO-N+1}}, g(x,y,t_{\text{LOTO-Time-(N+1)}}) \right) \right] dS$.

The informativeness of *Block $N+1$* is evaluated by measuring the relative change in effective risk upon its exclusion, indicating whether the block contributes valuable information or introduces noise. This change is captured by using the following:

$$\beta_{N+1} = \frac{R_{\text{eff},k}^{N+1} - R_{\text{eff},k}^{\text{LOTO-Time-}N+1}}{|R_{\text{eff},k}^{N+1}| + \epsilon},$$

where $R_{\text{full}}^{\text{eff},k}$ is the effective risk with learning from all blocks, including *Block $N+1$*, $R_{\text{LOTO-Time-(N+1)}}^{\text{eff},k}$ is the effective risk when *Block $N+1$* is excluded, and $\epsilon$ is a small positive constant to avoid division by zero:

- If $\beta_{N+1} > 0$: Removing *Block $N+1$* reduces risk indicating noisy or harmful.

- If $\beta_{N+1} < 0$: Removing *Block $N+1$* increases risk indicating informative or beneficial.

Repeat the LOTO procedure for each block $i \in \{1, 2, \ldots, N\}$ to compute a complete timewise risk profile. Compute the mean and standard deviation as $\beta_{N+1}^{\text{norm}} = \frac{\beta_{N+1} - \mu_\beta}{\sigma_\beta + \epsilon}$ and $\sigma_\beta = \sqrt{\frac{1}{N-1} \sum_{k=1}^{N} (\beta_k - \mu_\beta)^2}$, where $\mu_\beta = \frac{1}{N-1} \sum_{k=1}^{N} \beta_k$. This normalization step ensures the values are standardized and scale-invariant. The process of determining the adaptive weight for each block is then as follows:

$$\beta^t = \left[ \exp\left( -\beta_{N+1}^{\text{norm}} / \tau \right) \right]^\lambda,$$

where the $\tau$ and $\lambda$ controls the adaptive weights of the distribution.

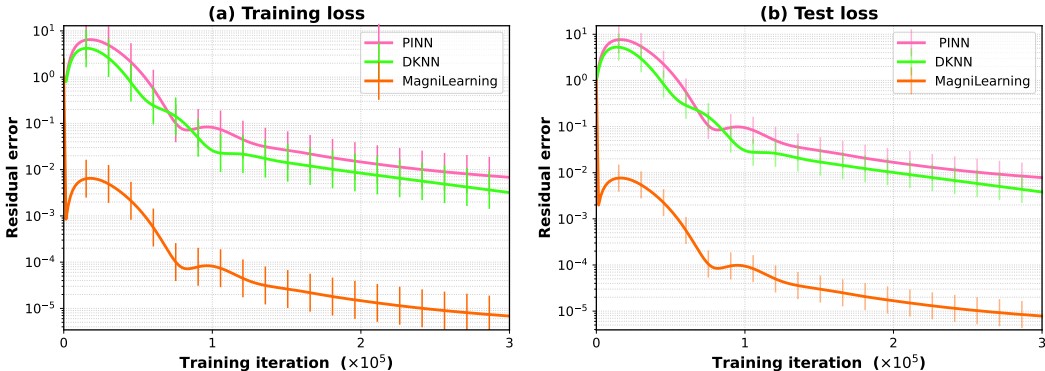

Figure 1: Comparison of training and test losses for PINN, DKNN, and MagniLearning on the Uniformly Distributed Axial Load on a Rod problem. MagniLearning consistently achieves lower losses than the baselines, highlighting its ability to prioritize informative data and knowledge and improve convergence stability. Results are averaged over 20 independent trials.

**Definition 3.5** Define the effective label for the $(N + 1)$-th smooth set as:

$$y_{\text{eff},k}^{\text{Mag}} = \arg\min_y \left\{ \alpha^t \sum_{t_{i-1}}^{t_i} \beta^t \int_S \left[ \mu(x,y)r(h(x,y,t)f(x,y,t)) \right. \right.$$

$$\left. \left. + (1 - \mu(x,y))r_k(h(x,y,t), g(x,y,t)) \right] dS \, dt \right\}.$$

The optimal effective risk is: $r_{\text{eff},k}^{\text{Mag}} = \alpha^t \sum_{t_{i-1}}^{t_i} \beta^t \int_S \left[ \mu(x,y)r(y_{\text{eff},k}^{\text{Mag}}, f(x,y,t)) + (1 - \mu(x,y))r_k(y_{\text{eff},k}^{\text{Mag}}, g(x,y,t)) \right] dS \, dt$, where $\alpha^t = \frac{1}{\sum_j \left[ \exp(\beta_j^{\text{norm}}/\tau) \right]^\chi}$.

### 3.4 Weighted Loss Function

For each smooth region and time block, we compute a weighted risk that captures both spatial and temporal contributions to the learning objective. In the discrete setting, this is obtained by summing the loss values over all training samples within the given region–time interval, scaled by their corresponding adaptive weights. This formulation serves as the basis for combining all components into the final weighted loss function.

**Definition 3.6.** The optimal effective risk for the **MagniLearning** that combines LORO, LOTO, and LOKO is: $r_{\text{eff},k}^{\text{Mag}} = \alpha_t \sum_t \beta_t \gamma_i \sum_{i \in I_{\phi,k}} \Gamma_i \left[ \mu_i r(h(x_i, y_i, t), f(x_i, y_i, t)) + (1 - \mu_i) \cdot r_k(h(x_i, y_i, t), g(x_i, y_i, t)) \right]$.

## 4 Experiments

In this paper, we conduct comparative studies on four distinct beam problems. Specifically, we benchmark our proposed MagniLearning method, which integrates LORO, LOTO, and LOKO, against the standard Physics-Informed Neural Network (PINN) Raissi et al. (2017) and the Domain Knowledge Neural Network (DKNN) Yang & Ren (2022). The objective is to evaluate the predictive accuracy and stability of MagniLearning in comparison with the baseline PINN and DKNN approaches.

We use a two-layer MLP with 50 units per layer and Tanh activation, initialized with Xavier initialization. Training is conducted using the Adam optimizer (learning rate $10^{-2}$) for 2000 epochs. For PINN, the objective combines the mean-squared PDE residual with boundary-condition penalties, while DKNN augments this objective with a conservative box prior weighted by $\lambda = 0.9$. MagniLearning further introduces adaptive reweighting with $\beta = 0.5$ and $\lambda = 0.5$. We report both training and test residual errors, and present convergence results on a logarithmic scale.

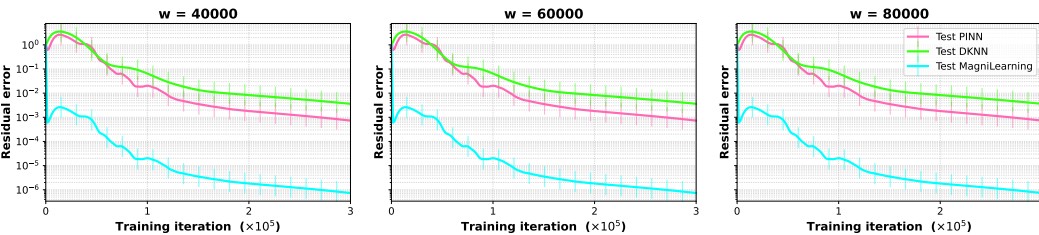

Figure 2: MagniLearning consistently outperforms PINN and DKNN across different loading scenarios with $w \in \{40{,}000, 60{,}000, 80{,}000\}$ N/m on the Uniformly Distributed Axial Load on a Rod problem, achieving faster convergence and lower residuals.

Figure 1 highlights the effectiveness of MagniLearning on the first problem, *Uniformly Distributed Axial Load on a Rod*. Compared with PINN and DKNN, MagniLearning converges faster and achieves lower training and testing errors, underscoring the strength of its adaptive weighting strategy. To further assess generalization, we evaluate the same benchmark under different loading/boundary conditions with $w = 40{,}000$, $60{,}000$, and $80{,}000$. The corresponding training losses, shown in Figure 2, confirm that MagniLearning consistently achieves both faster convergence and higher accuracy across all tested cases.

These results highlight a key distinction between MagniLearning and traditional approaches. PINN and DKNN directly minimize the residuals of governing equations and boundary conditions, which often leads to training instabilities, particularly near sharp gradients or complex boundary regions. In contrast, MagniLearning introduces an adaptive weighting mechanism that dynamically balances the contributions of PDE residuals, boundary conditions, and domain knowledge constraints. By reweighting in response to local training difficulty, the method focuses learning capacity on challenging regions without overfitting to simpler or already well-satisfied constraints.

This adaptivity not only stabilizes training but also improves the efficiency of optimization by reducing wasted effort on redundant or low-impact residuals. As a result, MagniLearning demonstrates greater robustness to variations in problem setup, boundary conditions, and parameter scales, while maintaining reliable convergence behavior. These findings suggest that adaptive weighting provides a principled pathway toward more scalable and generalizable neural PDE solvers.

## 5 CONCLUSION

In this work, we presented *MagniLearning*, a unified adaptive weighting framework for neural PDE solvers. By dynamically quantifying the contributions of spatial regions, temporal segments, and knowledge, the method accelerates convergence and improves robustness against training instabilities. We further introduced an optimal risk objective that combines region-, time-, and knowledge-dependent weights under balanced normalization, and established new generalization guarantees supported by theoretical risk bounds.

Extensive experiments on four benchmark beam problems with diverse boundary conditions and loading scenarios demonstrated that MagniLearning consistently outperforms both PINN and DKNN in accuracy, stability, and efficiency. These results not only validate the theoretical analysis in practice but also highlight the promise of adaptive weighting as a principled direction for improving the reliability of physics-informed learning.

## 6 ETHICS STATEMENT

This manuscript benefited from the use of a large language model (ChatGPT) to aid with grammar polishing and clarity of expression. The authors made all substantive contributions, including conceptualization, methodology, analysis, and validation.

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

# A    Appendix: Theoretical Analysis

## Preliminaries and Setup

**Proposition A.1** *If*   $0 <$    $r_{eff,k} < r_{eff,k}^{LORO}$,    *then*   $\Gamma_k > 1$, *where* $\beta > 0$, $\lambda > 0$, *and* $\epsilon > 0$.

**Proof:** *Define* $\Delta_k := \frac{r_{eff,k} - r_{eff,k}^{LORO}}{|r_{eff,k}| + \epsilon}$, *where* $\Gamma_k = [\exp(-\beta \cdot \Delta_k)]^{\lambda}$.

*If* $0 < r_{eff,k} < r_{eff,k}^{LORO}$, *then* $\Delta_k < 0$, *so* $-\beta \cdot \Delta_k > 0$, *and hence* $\exp(-\beta \cdot \Delta_k) > 1$, *implying* $\Gamma_k > 1$.

**Proposition A.2** *Let* $\Gamma_1, \Gamma_2, \ldots, \Gamma_K \in \mathbb{R}_{>0}$ *be the weights, with* $K \geq 2$, *then the following holds:*

$$0 < \frac{\Gamma_i}{\sum_{j=1}^{K} \Gamma_j} < 1$$

**Proof.** *Since* $\Gamma_j > 0$ *for all* $j = 1, \ldots, K$, *it follows that the total sum is strictly positive:*

$$\sum_{j=1}^{K} \Gamma_j > 0 \quad \Rightarrow \quad \frac{1}{\sum_{j=1}^{K} \Gamma_j} > 0,$$

*which establishes the lower bound. Since* $K \geq 2$ *and each* $\Gamma_j > 0$, *the total sum strictly exceeds any individual* $\Gamma_k$:

$$\sum_{j=1}^{K} \Gamma_j > \Gamma_k > 0, \quad \forall k \in \{1, \ldots, K\}.$$

*By dividing* $\sum_{j=1}^{K} \Gamma_j$:

$$\frac{\Gamma_i}{\sum_{j=1}^{K} \Gamma_j} < 1 \quad \text{for all } i \text{ in} \quad [1, k],$$

*combining both bounds:*

$$0 < \frac{\Gamma_i}{\sum_{j=1}^{K} \Gamma_j} < 1.$$

**Proposition A.3** *Let* $K \geq 2$ *and let* $\Gamma_1, \ldots, \Gamma_K > 0$. *For any fixed* $i \in \{1, \ldots, K\}$,

$$0 < 1 - \frac{\Gamma_i}{\sum_{j=1}^{K} \Gamma_j} < 1.$$

**Proof.** Set $S := \sum_{j=1}^{K} \Gamma_j$. Then:

$$1 - \frac{\Gamma_i}{\sum_{j=1}^{K} \Gamma_j} = 1 - \frac{\Gamma_i}{S} = \frac{S - \Gamma_i}{S}.$$

*Lower bound:* Since $K \geq 2$ and all $\Gamma_j > 0$, we have:

$$S = \Gamma_i + \sum_{j \neq i} \Gamma_j \quad \text{with} \quad \sum_{j \neq i} \Gamma_j > 0,$$

hence $S - \Gamma_i = \sum_{j \neq i} \Gamma_j > 0$ and $S > 0$. Therefore:

$$\frac{S - \Gamma_i}{S} > 0.$$

*Upper bound:* Because $S - \Gamma_i < S$ and $S > 0$, we get:

$$\frac{S - \Gamma_i}{S} < \frac{S}{S} = 1.$$

Combining the two bounds yields:

$$0 < 1 - \frac{\Gamma_i}{\sum_{j=1}^{K} \Gamma_j} < 1.$$

**Proposition A.4** *Let* $K \geq 2$ *and* $\Gamma_1, \ldots, \Gamma_K > 0$. *Put:*

$$S := \sum_{j=1}^{K} \Gamma_j \quad and \ define \quad w_i := 1 - \frac{\Gamma_i}{S}, \quad i = 1, \ldots, n.$$

*Then* $0 < w_i < 1$. *If* $f_i(x) \geq 0$ *for all* $i$, *we have:*

$$0 \ \leq \ \sum_{i=1}^{n} w_i f_i(x) \ \leq \ \sum_{i=1}^{n} f_i(x).$$

**Proof.** Since $\Gamma_j > 0$ and $K \geq 2$, we have $0 < w_i < 1$ for each $i$. Thus, for every $i$:

$$0 \leq w_i f_i(x) \leq f_i(x) \qquad \big(\text{because } w_i \in (0,1) \text{ and } f_i(x) \geq 0\big).$$

Summing over $i = 1, \ldots, n$ yields:

$$0 \leq \sum_{i=1}^{n} w_i f_i(x) \leq \sum_{i=1}^{n} f_i(x).$$

Moreover, since each $w_i < 1$, the upper inequality is strict if at least one $f_i(x) > 0$; equality holds iff $f_i(x) = 0$ for all $i$.

## MAIN THEORETICAL RESULTS

The following theorem is adapted from Theorem 1 in Yang & Ren (2022) and Allen-Zhu et al. (2019) with modifications to fit our analysis.

**Theorem A.5** *Assume the network width satisfies* $m \geq \Omega\left(\phi^{-11}b^{-4}L^{15}d\rho^{-4}\bar{\lambda}^{-4}\alpha^{-4}\log^3(m)\right)$, *and the step size is set as* $\eta = O\left(\frac{d}{L^2 m}\right)$. *For any* $\epsilon > 0$ *and* $\phi \leq \tilde{O}\left(\epsilon L^{-9/2}\log^{-3}(m)\right)$, *with probability at least* $1 - O(\phi)$, *gradient descent achieves the following after* $T = O\left(\frac{L^2}{\phi^{1+2b}\rho\lambda\alpha}\log\left(\frac{1}{\epsilon\log(1/\phi)}\right)\right)$ *steps, the informed risk is bounded as:*

$$\widehat{R}_I(W^T) - R_{eff}^{LORO} \leq O(\epsilon),$$

*where* $\widehat{R}_{eff}^{LORO} = \sum_{k=1}^{N} r_{eff,k}^{LORO}$. *Additionally, the trained network output satisfies the proximity bound:*

$$\sum_{x_i \in S_z \cup S_g} (\mu_i + \lambda_i)\|h_{W^T,i} - y_{eff}^{LORO}\|^2 \leq O(\epsilon),$$

*where* $k(x_i)$ *is the index of the smooth set that includes* $x_i$, $\mu_i = 1 - \lambda\sum_z \mathbf{1}(x_i \in S_z)$, *and* $\lambda_i = \lambda\sum_g \mathbf{1}(x_i \in S_g)$.

In our generalization analysis, we redefine the hypothesis class as:

$$\mathcal{F} = \{r(h(x,y), \hat{y}) : h \in \mathcal{H}_{\gamma,p,q}\},$$

where $\mathcal{H}_{\gamma,p,q}$ denotes the class of depth-$H$ ReLU networks with weight matrices $\{W_\ell\}_{\ell=1}^{H}$ satisfying:

$$\prod_{\ell=1}^{H} \|W_\ell\|_{p\to q} \leq \gamma.$$

We assume that inputs are bounded as $\|(x_i, y_i)\|_{p^*} \leq B$ (with $\frac{1}{p} + \frac{1}{p^*} = 1$), and the loss $r(\cdot, \hat{y})$ is $L_r$-Lipschitz in its first argument, and that standard initialization/smoothness conditions hold so that $\gamma$ remains bounded during training. The hypothesis class is defined as the norm-bounded family of depth-H ReLU networks, given by $\mathcal{H}_{\gamma,p,q} = \left\{h : \text{depth-}H \text{ ReLU network with } \prod_{\ell=1}^{H}\|W_\ell\|_{p\to q} \leq \gamma\right\}$.

**Lemma A.6 (Norm-based Complexity)** *Neyshabur et al. (2015) Let $\mathcal{H}_{\gamma,p,q}$ denote the class of depth-H ReLU networks with weight matrices $\{W_\ell\}_{\ell=1}^H$ satisfying:*

$$\prod_{\ell=1}^H \|W_\ell\|_{p \to q} \leq \gamma.$$

*Assume that: (i) inputs are bounded as $\|(x_i, y_i)\|_{p^*} \leq B$ (with $1/p + 1/p^* = 1$); (ii) The loss $r(\cdot, \hat{y})$ is $L_r$-Lipschitz in its first argument; (iii) standard initialization/smoothness conditions hold so that $\gamma$ remains bounded during training. Then for any $0 < \delta < 1$, with probability at least $1 - \delta$ over a sample $S$ of size $n$:*

$$\mathfrak{R}_S\left(N_{\mu_{p,q}}^{d,H,\sigma_{ReLu}}\right) \leq \frac{\Phi'}{\sqrt{n}},$$

*where*

$$\Phi' = \sqrt{\gamma^2 \left(2H^{[\frac{1}{p^*} - \frac{1}{q}]+}\right)^{2(d-1)} \min\{p^*, 4\log(2D)\} \max_i \|x_i\|_{p^*}^2}$$

*where the second inequalities hold only if $1 \leq p \leq 2$, $\mathcal{R}_{m,p,D}^{linear}$ is the Rademacher complexity of D-dimensional linear predictors with unit $\ell_p$ norm with respect to a set of $m$ samples, and $p^*$ is such that $\frac{1}{p^*} + \frac{1}{p} = 1$.*

Compared to margin-based bounds Yang & Ren (2022), this norm-based formulation is advantageous because it links generalization directly to spectral/Frobenius norm growth, which can be explicitly tracked during training. It also avoids margin-related dependencies, making the analysis compatible with modern over-parameterized regimes where margins may vanish or be difficult to estimate.

Assume that for any smooth set $k \in \mathcal{U}_\phi(S_z)$ (containing at least one labeled sample), the effective label $y_{\text{eff},k}$ equivalently minimizes:

$$\sum_{i \in I_{\phi,k}} \left(\frac{1 - \beta}{n_z} \mathbf{1}(x_i \in S_z) r(h, f_i) + \frac{\beta}{n_g^0} \mathbf{1}(x_i \in S_g^0) r_K(h, g_i)\right),$$

and for any smooth set $k \in [N] \setminus \mathcal{U}_\phi(S_z)$ (not containing a labeled sample), $y_{\text{eff},k}$ equivalently minimizes:

$$\sum_{i \in I_{\phi,k}} \frac{1}{n_g''} \cdot r_K(h, g_i).$$

Letting $h_K^*$ and $h_{R,\beta}^*$ ( abbreviated as $h^*$) be the optimal hypotheses for the empirical risks, with probability at least $1 - O(\phi) - \delta$, for $\delta \in (0, 1)$ over the randomness of $W(0)$:

$$\frac{1}{n_g''} \sum_{x_i \in S_g''} \|h^* - y_{\text{eff},k(x_i,y_i)}\|^2 \leq \widetilde{\mathcal{O}}\left(\frac{L^{5/4} \phi^{1/2} \log^{1/4}(m)}{\sqrt{n}}\right),$$

and

$$\frac{1 - \beta}{n_z} \sum_{x_i \in S_z} \|h^* - y_{\text{eff},k(x_i,y_i)}\|^2 + \frac{\beta}{n_g'} \sum_{x_i \in S_g'} \|h^* - y_{\text{eff},k(x_i,y_i)}\|^2 \leq \widetilde{\mathcal{O}}\left(\frac{L^{5/4} \phi^{1/2} \log^{1/4}(m)}{\sqrt{n}}\right),$$

where $\widetilde{\mathcal{O}}\left(\frac{L^{5/4} \phi^{1/2} \log^{1/4}(m)}{\sqrt{n}}\right) = \mathcal{O}\left(\frac{L^{5/4} \phi^{1/2} \log^{1/4}(m)}{\sqrt{n}} (1/\phi) \log^{1/4}(m)\right).$

Let $\Gamma_i$ denote the normalized group indicator, defined as $\Gamma_i' = \frac{\Gamma_i}{\sum_i \Gamma_i}$. We apply Rademacher complexity version Mohri et al. (2018).

PROOF OF THEOREM 3.1.

The population risk of the trained hypothesis $h_{W^T,i}$ is bounded with probability at least $1 - O(\phi) - \delta$, for $\delta \in (0, 1)$, as follows:

$$R(W^T) \leq \frac{1}{n_s} \sum_{S_g \cup S_z} (1 - \frac{(\Gamma_i)}{\sum_i \Gamma_i}) r(h_{W^T,i}, \hat{y}_i) + \frac{1}{n_s} \sum_{S_g \cup S_z} \frac{\Gamma_i}{\sum_i \Gamma_i} r(h_{W^T,i}, \hat{y}_i)$$

$$+ O\left( \frac{\Phi'}{\sqrt{n}} + 3\sqrt{\frac{\log \frac{2}{\delta}}{2n}} \right) \tag{5}$$

$$\leq \frac{1}{n_s} \sum_{S_g \cup S_z} (1 - \Gamma_i') r(h_{W^T,i}, \hat{y}_i) + \frac{1}{n_s} \sum_{S_g \cup S_z} \Gamma_i' r(h_{W^T,i}, \hat{y}_i)$$

$$+ O\left( \frac{\Phi'}{\sqrt{n}} + 3\sqrt{\frac{\log \frac{2}{\delta}}{2n}} \right), \tag{6}$$

where $\Phi' = \sqrt{\gamma^2 \left( 2H^{[\frac{1}{p^*} - \frac{1}{q}]+} \right)^{2(d-1)} \min\{p^*, 4\log(2D)\} \max_i \|x_i\|_{p^*}^2}$

Following Proposition A.2, we have $\frac{\Gamma_i}{\sum_i \Gamma_i} \leq 1$ and put $1 - \Gamma = \max \left\{ 1 - \Gamma_i' \mid \text{for all } i \right\}$.

**Notation.** Let $u_i := h_{W^{(T)}}$ and $v_i := y_{\text{eff},k}(x_i, y_i)$. Assume the loss $r(\cdot, \hat{y}^i) : \mathbb{R}^m \to \mathbb{R}$ is differentiable in its first argument on a neighborhood of the line segment $\{ v_i + t(u_i - v_i) : t \in [0, 1] \}$. Then, by applying the (univariate) Mean Value Theorem to $\phi_i(t) = r(v_i + t(u_i - v_i), \hat{y}^i)$, there exists some $t_i \in (0, 1)$ and $\xi_i = v_i + t_i(u_i - v_i)$.

$$r(h_{W^{(T)},i}, \hat{y}_i) - r(y^{\text{eff},k}(x_i), \hat{y}_i) = \nabla r(\xi_i, \hat{y})^\top \left( h_{W^{(T)},i} - y^{\text{eff},k}(x_i) \right).$$

Taking the absolute value of both sides and applying the Cauchy–Schwarz inequality yields:

$$\left\| r(h_{W^T,i}, \hat{y}_i) \right\| - \left\| r(y_{\text{eff},k(x_i,y_i)}, \hat{y}_i) \right\| \leq \left\| r(h_{W^T,i}, \hat{y}_i) - r(y_{\text{eff},k(x_i,y_i)}, \hat{y}_i) \right\|$$
$$\leq \left\| \nabla r(\xi_i) \right\| \cdot \left\| h_{W^T,i} - y_{\text{eff},k(x_i,y_i)} \right\|. \tag{7}$$

Assume there is $C$ such that $\|\nabla r(\xi_i)\| \leq C$, where $C$ is a bounded constant (e.g., if $r$ is smooth or Lipschitz with respect to the first argument).

We express the population risk as a weighted sum over the dataset $S$:

$$\frac{1}{n_s} \sum_{S_g \cup S_z} (1 - \Gamma_i') r(h_{W^T,i}, \hat{y}_i) + \frac{1}{n_s} \sum_{S_g \cup S_z} \Gamma_i' r(h_{W^T,i}, \hat{y}_i)$$

$$\leq \frac{1}{n_s} \sum_{S_g \cup S_z} (1 - \Gamma_i') \left[ r(y_{\text{eff},k(x_i,y_i)}, \hat{y}_i) + \|\nabla r(\xi_i)\| . \|h_{W^T,i} - y_{\text{eff},k(x_i,y_i)}\| \right]$$

$$+ \frac{1}{n_s} \sum_{S_g \cup S_z} \Gamma_i' \left( \left[ r(y_{\text{eff},k(x_i,y_i)}, \hat{y}_i) + \|\nabla r(\xi_i)\| \|h_{W^T,i} - y_{\text{eff},k(x_i,y_i)}\| \right] \right)$$

$$= \frac{1}{n_s} \sum_{S_g \cup S_z} (1 - \Gamma_i') r(y_{\text{eff},k(x_i,y_i)}, \hat{y}_i) + \frac{1}{n_s} \sum_{S_g \cup S_z} \Gamma_i \, r(y_{\text{eff},k(x_i,y_i)}, \hat{y}_i) + O(\varepsilon)$$

By applying Theorem A.3, we have:

$$\frac{1}{n_s} \sum_{S_g \cup S_z} (1 - \Gamma_i^{'}) r(y_{\text{eff},k(x_i,y_i)}, \hat{y}_i) + \frac{1}{n_s} \sum_{S_g \cup S_z} \Gamma_i^{'} r(y_{\text{eff},k(x_i,y_i)}, \hat{y}_i)$$

$$\leq \frac{1}{n_s} \sum_{S_g \cup S_z} (1 - \Gamma_i^{'}) \left[ r(h^*, \hat{y}_i) + \|\nabla r(\xi_i, y_i)\| \cdot \|h^* - y_{\text{eff},k(x_i,y_i)}\| \right]$$

$$+ \frac{1}{n_s} \sum_{S_g \cup S_z} \Gamma_i^{'} \left[ r(h^*, \hat{y}_i) + \|\nabla r(\xi_i)\| \cdot \|h^* - y_{\text{eff},k(x_i,y_i)}\| \right]$$

$$\leq (1 - \Gamma) \frac{1}{n_s} \sum_{S_g \cup S_z} r(h^*, \hat{y}_i) + (1 - \Gamma_i^{'}) \|\nabla r(\xi_i)\| \cdot \frac{1}{n_s} \sum_{S_g \cup S_z} \|h^* - y_{\text{eff},k(x_i,y_i)}\|$$

$$+ \frac{1}{n_s} \sum_{S_g \cup S_z} \Gamma_i^{'} r(h^*, \hat{y}_i) + \Gamma_i^{'} \|\nabla r(\xi_i)\| \frac{1}{n_s} \sum_{S_g \cup S_z} \|h^* - y_{\text{eff},k(x_i,y_i)}\|$$

$$= (1 - \Gamma) \frac{1}{n_s} \sum_{S_g \cup S_z} r(h^*, \hat{y}_i) + \|\nabla r(\xi_i, y_i)\| \frac{1}{n_s} \sum_{S_g \cup S_z} \|h^* - y_{\text{eff},k(x_i,y_i)}\|$$

$$+ \frac{1}{n_s} \sum_{S_g \cup S_z} \Gamma_i^{'} r(h^*, \hat{y}_i)$$

$$\leq (1 - \Gamma) \frac{1}{n_s} \sum_{S_g \cup S_z} r(h^*, \hat{y}_i) + \frac{1}{n_s} \sum_{S_g \cup S_z} \Gamma_i^{'} r(h^*, \hat{y}_i) + \mathcal{O}\left( \frac{L^{5/4} \phi^{1/2} \log^{1/4}(m)}{\sqrt{n}} \right)$$

Finally, by applying A.3 and A.4, we conclude that:

$$R(W^T) \leq (1 - \Gamma) \frac{1}{n_s} \sum_{S_g \cup S_z} r(h^*, \hat{y}_i)$$

$$+ \frac{1}{n_s} \sum_{S_g \cup S_z} \Gamma_i^{'} r(h^*, \hat{y}_i)$$

$$+ O(\frac{\Phi'}{\sqrt{n}} + 3\sqrt{\frac{\log \frac{2}{\delta}}{2n}}) \tag{8}$$

$$+ \widetilde{\mathcal{O}}\left( \frac{L^{5/8} \phi^{1/4} \log^{1/8}(m)}{\sqrt{n}} \right) + O(\varepsilon) \tag{9}$$

$$\leq O(\sqrt{\varepsilon}) + \widetilde{\mathcal{O}}\left( \frac{L^{5/4} \phi^{1/2} \log^{1/4}(m)}{\sqrt{n}} \right) + O(\varepsilon)$$

$$+ \frac{1}{n_s} \sum_{(x_i,y_i) \in I_{\phi,k}} \Gamma_i^{'} r(h^*, \hat{y}_i)$$

$$+ O(\frac{\Phi'}{\sqrt{n}} + 3\sqrt{\frac{\log \frac{2}{\delta}}{2n}}). \tag{10}$$

We absorb $\frac{\lambda}{n_g} \sum_{S_z \cup S_g} r(h_i^*, y_i)$ into $O(\sqrt{\epsilon})$ because the risk functions are upper bounded, and $S_z \cup S_g$ is the set of samples sharing same smooth sets with $S_z$. Therefore:

$$\frac{1}{n_g} \sum_{S_z \cup S_g} r(h^*, \hat{y}_i) \leq O\left( \frac{n_g^{'}}{n_g} \right) \leq O(n_z \phi_b) \leq O(\sqrt{\epsilon}).$$

By replacing A.5, we have:

$$R(W^T) \leq \frac{1}{n_s} \sum_{S_g \cup S_z} \Gamma_i \, r(h^*, \hat{y}_i)$$

$$+ O(\varepsilon) + \mathcal{O}\left( \frac{L^{5/8} \phi^{1/4} \log^{1/8}(m)}{\sqrt{n}} \right)$$

$$+ O\left( \frac{\Phi'}{\sqrt{n}} + 3\sqrt{\frac{\log \frac{2}{\delta}}{2n}}. \right) + O(\sqrt{\varepsilon})$$

$$= \frac{1}{n_s} \sum_{S_g \cup S_z} \Gamma_i \, r(h^*, \hat{y}_i) + O(\varepsilon) + \mathcal{O}\left( \frac{L^{5/8} \phi^{1/4} \log^{1/8}(m)}{\sqrt{n}} \right)$$

$$+ O\left( \frac{\Phi'}{\sqrt{n}} + 3\sqrt{\frac{\log \frac{2}{\delta}}{2n}}. \right) + O(\sqrt{\varepsilon})$$

$$\leq \frac{1}{n_s} \sum_{S_g \cup S_z} r(h^*, \hat{y}_i) + O(\sqrt{\varepsilon}) + \mathcal{O}\left( \frac{L^{5/8} \phi^{1/4} \log^{1/8}(m)}{\sqrt{n}} \right) \quad )$$

$$+ O\left( \frac{\Phi'}{\sqrt{n}} + 3\sqrt{\frac{\log \frac{2}{\delta}}{2n}}. \right).$$

where it is trivial that $1 - \Gamma \leq 1$.

PROOF OF THEOREM 3.2.

By the generalization bound using Lemma A.5, the population risk is bounded with probability at least $1 - \delta$, where $\delta \in (0, 1)$, as follows:

$$R(W^{(T)}) = (1 - \kappa)R(W^{(T)}) + \kappa R(W^{(T)})$$

$$\leq \frac{1 - \kappa}{n_z} \sum_{(x_i, y_i) \in S_z} r(h_{W^{(T)}, i}, \hat{y}_i) + \frac{\kappa}{n_g''} \sum_{(x_i, y_i) \in S_g''} r(h_{W^{(T)}, i}, \hat{y}_i)$$

$$+ O(\epsilon) + \frac{1}{n} \mathcal{O}\left( \frac{L^{5/4} \phi^{1/2} \log^{1/4}(m)}{\sqrt{n}} \right) + O\left( \frac{\Phi'}{\sqrt{n}} + 3\sqrt{\frac{\log \frac{2}{\delta}}{2n}} \right)$$

$$\leq \frac{1 - \kappa}{n_z} \sum_{(x_i, y_i) \in S_z} r(h^*, \hat{y}_i) + \frac{\kappa}{n_g''} \sum_{(x_i, y_i) \in S_g''} r(h^*, \hat{y}_i)$$

$$+ O(\sqrt{\epsilon}) + \frac{1}{n} \cdot \mathcal{O}\left( \frac{L^{5/4} \phi^{1/2} \log^{1/4}(m)}{\sqrt{n}} \right) + O\left( \frac{\Phi'}{\sqrt{n}} + 3\sqrt{\frac{\log \frac{2}{\delta}}{2n}} \right).$$

where $\Phi' = \sqrt{\gamma^2 \left( 2H^{[\frac{1}{p^*} - \frac{1}{q}]+} \right)^{2(d-1)} \min\{p^*, 4\log(2D)\} \max_i \|x_i\|_{p^*}^2}$

By applying proof in Theorem 4.2, Theorem A.3, and the same process in the previous proof, we have:

$$R(W^{(T)}) = (1 - \kappa)R(W^{(T)}) + \kappa R(W^{(T)})$$

$$\leq O(\sqrt{\epsilon}) + (1 - \kappa) \widehat{R}_S^\kappa + \kappa \widehat{R}_{S''}^\kappa \tag{11}$$

$$+ \mathcal{O}\left( \frac{L^{5/4} \phi^{1/2} \log^{1/4}(m)}{\sqrt{n}} \right) + O\left( \frac{\Phi'}{\sqrt{n}} + 3\sqrt{\frac{\log \frac{2}{\delta}}{2n}} \right) \tag{12}$$

where $\Phi' = CL_r \gamma \left( 2H \left( \frac{1}{p^*} - \frac{1}{q} \right) + (d - 1) \right) \min\{p^*, 4\log(2D)\}.$

We follow Yang et al. (2022) and keep the effect of labeled and confident data unchanged. We adjust the weight of unlabeled data ($S''_g$) using a dynamic factor $\kappa_i$. This means $\kappa$ is applied only to $S''_g$:

$$\widehat{R}^\kappa_{I,G}(W) = (1-\beta)\frac{1}{n_z}\sum_{i\in S_z} r\big(h(x_i,y_i), f(x_i,y_i)\big)$$

$$+ \beta\frac{1}{n'_g}\sum_{i\in S'_g} r_K\big(h(x_i,y_i), g(x_i,y_i)\big) + \frac{1}{n''_g}\sum_{i\in S''_g}\frac{\kappa_i}{\sum_i \kappa_i} r_K\big(h(x_i,y_i), g(x_i,y_i)\big). \tag{13}$$

Let $\widehat{R}^\kappa_S = \frac{1}{n_z}\sum_{z\in S_z} r(h^*,\hat{y})$ where $h^* = \arg\min(1-\lambda)\frac{1}{n_z}\sum_{S_z} r(h(x_i,y_i),g(x_i,y_i)) + \lambda\frac{1}{n'_g}\sum_{S'_g} r_K(h(x_i,y_i),f(x_i,y_i))$ and $\widehat{R}^\kappa_{S''} = \frac{1}{n''_g}\sum_{i=1}^{n''_g} r_K(h^*,\hat{y})$ such that $h^*_K = \arg\min_h \frac{1}{n''_g}\sum_{x_i\in S''_g}\Gamma_i r_K(h(x_i,y_i),g(x_i,y_i))$.

**Theorem A.7** *Assume $\phi \leq \widetilde{O}(\epsilon^2 L^{-9/2}\log^{-3}(m))$ and $\Phi \leq (\epsilon/n)^{1/b}$ and let complexity requirement is given by:*

$$m \geq \Omega\left(\phi^{-1}b^{-4}L^{15}d\rho^{-4}\bar{\lambda}^{-4}\alpha^{-4}\log^3(m)\right), \tag{14}$$

*where $\bar{\lambda}$ is defined as: $\bar{\lambda} = \Omega\left(\min(1-\lambda,\lambda)\cdot\mathbf{1}_{\lambda\in(0,1)} + \mathbf{1}_{\lambda\in\{0,1\}}\right)$. The step size $\eta$ for gradient descent is set as: $\eta = \mathcal{O}\left(\frac{d}{L^2 m}\right)$, after $T$ iterations, where $T = \mathcal{O}\left(\frac{L^2}{\phi^{1+2b}\rho\bar{\lambda}\alpha}\log\left(\frac{1}{\epsilon}\log\left(\frac{1}{\phi}\right)\right)\right)$. Then, with probability at least $1-O(\phi)-\delta$, for $\delta\in(0,1)$, the population risk satisfies the following:*

$$R(W^{(T)}) \leq \mathcal{O}(\sqrt{\varepsilon}) + \widehat{R}_S + \widehat{R}^\kappa_{S''} + \mathcal{O}\left(\frac{L^{5/4}\phi^{1/2}\log^{1/4}(m)}{\sqrt{n}}\right) + O\left(\frac{\Phi'}{\sqrt{n}} + 3\sqrt{\frac{\log\frac{2}{\delta}}{2n}}\right) \tag{15}$$

*Where $\Phi' = \sqrt{\gamma^2\left(2H^{[\frac{1}{p^*}-\frac{1}{q}]+}\right)^{2(d-1)}\min\{p^*, 4\log(2D)\}\max_i\|x_i\|^2_{p^*}}$.*

**Proof.** The proof process follows the proofs of Theorem 1 and Theorem 2; therefore, we only provide the details for the parts that differ:

$$\frac{(1-\lambda)}{n_g}\sum_{S_g} r(h_{W^T,i},\hat{y}_i) + \frac{\lambda}{n_z}\sum_{S_z} r(h_{W^T,i},\hat{y}_i)$$

$$\leq \frac{(1-\lambda)}{n_z}\sum_{S_z} r(y_{\text{eff},k(x_i,y_i)},\hat{y}_i) + O(\varepsilon) + \frac{\lambda}{n_g}\sum_{S_g} r(y_{\text{eff},k(x_i,y_i)},\hat{y}_i)$$

$$\leq O(\varepsilon) + \frac{(1-\lambda)}{n_z}\sum_{S_z} r(y_{\text{eff},k(x_i,y_i)},\hat{y}_i) + \frac{\lambda}{n'_g}\sum_{S'_g} r(y_{\text{eff},k(x_i,y_i)},\hat{y}_i) + \frac{\lambda}{n''_g}\sum_{S''_g} r(y_{\text{eff},k(x_i,y_i)},\hat{y}_i)$$

$$\leq O(\varepsilon) + \frac{(1-\lambda)}{n_z}\sum_{S_z} r(h^*,\hat{y}_i) + \frac{\lambda}{n'_g}\sum_{S'_g} r(h^*,\hat{y}_i)$$

$$+ \frac{\lambda}{n''_g}\sum_{S''_g} r(h^*,\hat{y}_i) + \|\nabla r(\xi_i)\|\frac{(1-\lambda)}{n_z}\sum_{S_z}\|h^* - y_{\text{eff},k(x_i,y_i)}\|$$

$$+ \|\nabla r(\xi_i)\|\frac{\lambda}{n'_g}\sum_{S'_g}\|h^* - y_{\text{eff},k(x_i,y_i)}\| + \|\nabla r(\xi_i)\|\frac{\lambda}{n''_g}\sum_{S''_g}\|h^* - y_{\text{eff},k(x_i,y_i)}\|$$

$$\leq \frac{(1-\lambda)}{n_z}\sum_{S_z} r(h^*,\hat{y}_i) + \frac{\lambda}{n'_g}\sum_{S'_g} r(h^*,\hat{y}_i)$$

$$+ \frac{\lambda}{n''_g}\sum_{S''_g} r(h^*,\hat{y}_i) + \|\nabla r(\xi_i)\|\frac{1-\lambda}{n_z}\sum_{S_z}\|h^* - y_{\text{eff},k(x_i,y_i)}\|$$

$$+ \|\nabla r(\xi_i)\|\frac{\lambda}{n'_g}\sum_{S'_g}\|h^* - y_{\text{eff},k(x_i,y_i)}\| + \|\nabla r(\xi_i)\|\frac{\lambda}{n''_g}\sum_{S''_g}\|h^* - y_{\text{eff},k(x_i,y_i)}\|$$

Finally, we have:

$$R(W) \leq O(\varepsilon) + (1-\lambda)\frac{1}{n_z}\sum_{S_z} r(y_{\text{eff},k(x_i,y_i)},\hat{y}_i) + \lambda\frac{1}{n_g}\sum_{S'_g} r(y_{\text{eff},k(x_i,y_i)},\hat{y}_i)$$

$$+ \lambda\frac{1}{n_g}\sum_{S''_g}(1 - \frac{\kappa_i}{\sum_i \kappa_i})r(y_{\text{eff},k(x_i,y_i)},\hat{y}_i)$$

$$+ \lambda\frac{1}{n_g}\sum_{S''_g}\frac{\kappa_i}{\sum_i \kappa_i}r(y_{\text{eff},k(x_i,y_i)},\hat{y}_i) + O\left(\frac{\Phi'}{\sqrt{n}} + 3\sqrt{\frac{\log\frac{2}{\delta}}{2n}}\right)$$

$$+ \mathcal{O}\left(\frac{L^{5/4}\phi^{1/2}\log^{1/4}(m)}{\sqrt{n}}\right)$$

$$\leq \mathcal{O}(\sqrt{\varepsilon}) + \widehat{R}_S + \widehat{R}^\kappa_{S''}$$

$$+ \mathcal{O}\left(\frac{L^{5/4}\phi^{1/2}\log^{1/4}(m)}{\sqrt{n}}\right) + O\left(\frac{\Phi'}{\sqrt{n}} + 3\sqrt{\frac{\log\frac{2}{\delta}}{2n}}\right),$$

Where $\Phi' = \sqrt{\gamma^2 \left(2H^{[\frac{1}{p^*}-\frac{1}{q}]+}\right)^{2(d-1)} \min\{p^*, 4\log(2D)\} \max_i \|x_i\|^2_{p^*}}$. and $\lambda$, $(1-\frac{\kappa_i}{\sum_i \kappa})$, $\frac{1}{n_z}$, and $\frac{1}{n_z}$ are all less than 1.

# B  APPENDIX: NUMERICAL RESULTS

## B.1  PROBLEM 1: UNIFORMLY DISTRIBUTED AXIAL LOAD ON A ROD

The problem is a typical finite element problem that involves a rod subjected to a uniformly distributed axial load along its length. The objective is to model the horizontal deformation of the rod under the axial load using machine learning. In this scenario, the axial load is uniformly distributed along the length $L$ of the rod. Using Hooke's Law, the problem can be formulated in terms of a differential equation governing the rod's deformation. The governing equation for the displacement $u(x)$ is given by:

$$AE\frac{d^2u(x)}{dx^2} = -cx, \tag{16}$$

where $A$ is the cross-sectional area of the rod, $E$ is the Young's modulus of the material, and $c$ is a constant that represents the uniformly distributed load per unit length. Solving this differential equation with the appropriate boundary conditions, we get the *analytical solution*:

$$u(x) = \frac{c}{6AE}\left(-x^3 + 3L^3 x\right). \tag{17}$$

The rod is fixed at one end, leading to the following *boundary conditions*:

- $u(0) = 0$, indicating no displacement at the fixed end.

- $\left.\frac{du}{dx}\right|_{x=L} = 0$, meaning that the strain is zero at the free end.

The regular PINN solves this problem by directly minimizing the residual of the governing equation. The total loss function for the regular PINN is composed of the residual of the governing PDE and the enforcement of the boundary condition:

$$\mathcal{L}_{\text{PINN}}(x) = \text{MSE}\left(\frac{d^2u(x)}{dx^2} + 2x\right) + \text{MSE}(u(0)) + \text{MSE}\left(\left.\frac{du}{dx}\right|_{x=L}\right), \tag{18}$$

where MSE represents the mean squared error, and the residual loss measures the discrepancy between the second derivative of the neural network and the right-hand side of the governing equation, while the boundary terms penalize deviations from the imposed boundary conditions.

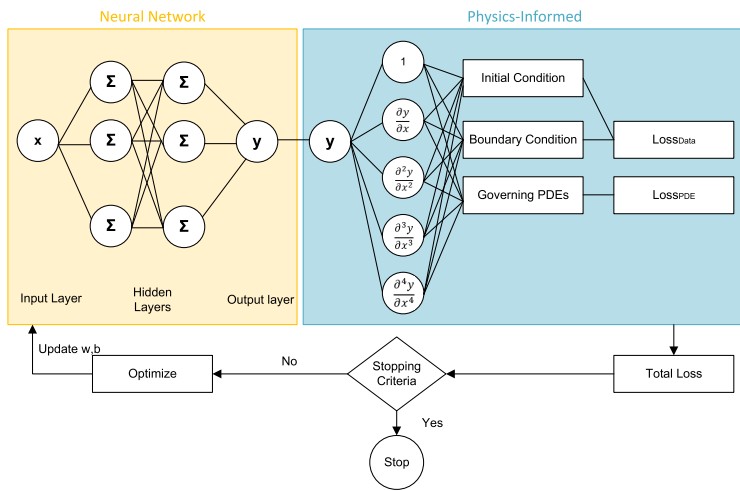

Figure 3: The architecture of PINN for benchmark solid structure PDE problems.

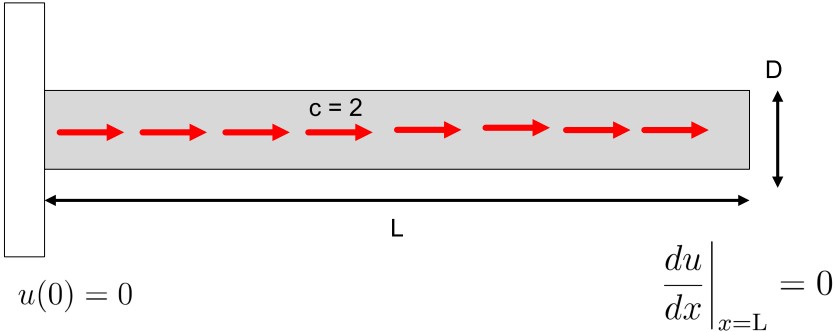

$$u(0) = 0$$

$$\frac{du}{dx}\bigg|_{x=\mathrm{L}} = 0$$

PDE:  $AE\dfrac{d^2 u}{dx^2} = -cx, \quad x \in (0, L)$

Analytical Solution:  $u(x) = \dfrac{c}{6AE}\left(-x^3 + 3L^3 x\right)$

Figure 4: Schematic representations of a Cantilever beam subjected to a point load at the free end (Problem 1).

Furthermore, incorporating domain knowledge into the PINN framework is used to improve the model's convergence and accuracy. In this case, upper and lower bounds are introduced for the displacement, which serve as domain knowledge constraints. The domain knowledge-enforced loss function modifies the regular PINN loss to include a term that penalizes the network's output if it deviates from physically plausible upper and lower bounds. Let $y(x)$ denote the predicted displacement by the network, and $u_{\text{lower}}(x)$ and $u_{\text{upper}}(x)$ denote the lower and upper bounds, respectively:

$$u_{\text{lower}}(x) = \cos(x), \quad u_{\text{upper}}(x) = \sin(x). \tag{19}$$

The domain knowledge loss ensures that the solution stays within these bounds:

$$\mathcal{L}_{\text{DKNN}}(x) = \text{MSE}\left(\frac{d^2 u(x)}{dx^2} - f(x)\right) + \text{MSE}(u(0)) + \text{MSE}\left(\frac{du}{dx}\bigg|_{x=L}\right)$$

$$+ \text{MSE}\left(u(x) - \text{Clamp}(u(x), u_{\text{lower}}, u_{\text{upper}})\right). \tag{20}$$

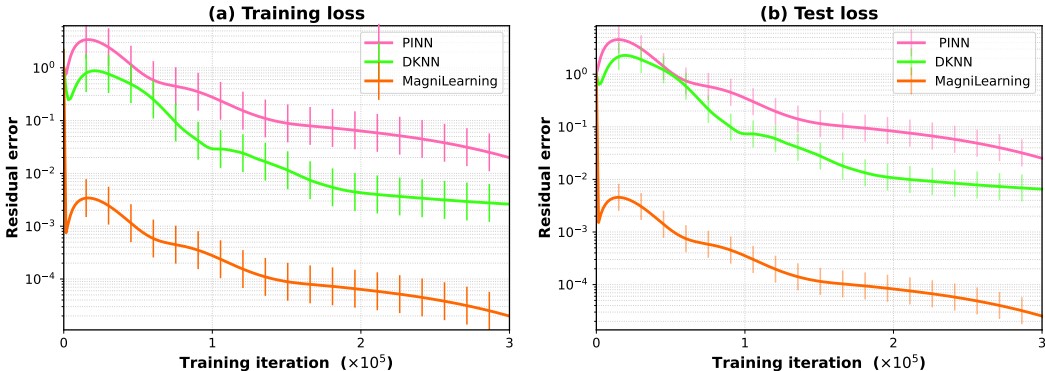

Figure 5: Comparison of training and test losses for PINN, DKNN, and MagniLearning on the Uniformly Distributed Axial Load on a Rod problem. MagniLearning consistently achieves lower losses than the baselines, highlighting its ability to prioritize informative data and knowledge and improve convergence stability. Results are averaged over 20 independent trials.

The total loss for the DKNN is the weighted combination of the physics-informed loss and the domain-knowledge loss:

$$\mathcal{L}_{\text{Total}} = \lambda \mathcal{L}_{\text{PINN}} + (1 - \lambda)\mathcal{L}_{\text{DKNN}} \tag{21}$$

where $\lambda$ is a hyperparameter that balances the importance of the loss informed by physics and the loss of domain knowledge.

Beam equations have previously been addressed using machine learning, with PINN and DKNN serving as standard benchmarks for structural PDEs. These methods approximate deflections and stresses under various loads and boundary conditions, but often face convergence difficulties and reduced accuracy near complex regions. Building on this line of work, we apply MagniLearning to this beam problems, and as illustrated in Figure 5, adaptive reweighting yields faster convergence and more accurate solutions compared to existing approaches.

### B.2  Problem 2: Simply Supported Beam with Uniformly Distributed Loading

The problem involves a simply supported beam subjected to a uniformly distributed load along its length. According to classical beam theory, the deflection of a beam subjected to bending moments and shear forces can be described by the following governing equation. The bending moment $M(x)$ is related to the deflection $y(x)$ as:

$$EI\frac{d^2y(x)}{dx^2} = M(x), \tag{22}$$

where $E$ is the Young's modulus of the beam, $I$ is the moment of inertia of the beam's cross-sectional area, and $M(x)$ is the internal bending moment. The relationship between the bending moment, the shear force $V$, and the distributed load $w(x)$ is given by:

$$\frac{dM}{dx} = V, \quad \frac{dV}{dx} = -w(x). \tag{23}$$

Substituting these into the second-order differential equation, we get a fourth-order differential equation for the deflection $y(x)$ under the distributed load $w(x)$:

$$EI\frac{d^4y(x)}{dx^4} = -w(x). \tag{24}$$

For a simply supported beam of length $L$ under uniform loading $w(x) = w$, the boundary conditions are:

$$y(0) = 0, \quad y(L) = 0, \frac{d^2y}{dx^2}\bigg|_{x=0} = 0, \quad \frac{d^2y}{dx^2}\bigg|_{x=L} = 0. \tag{25}$$

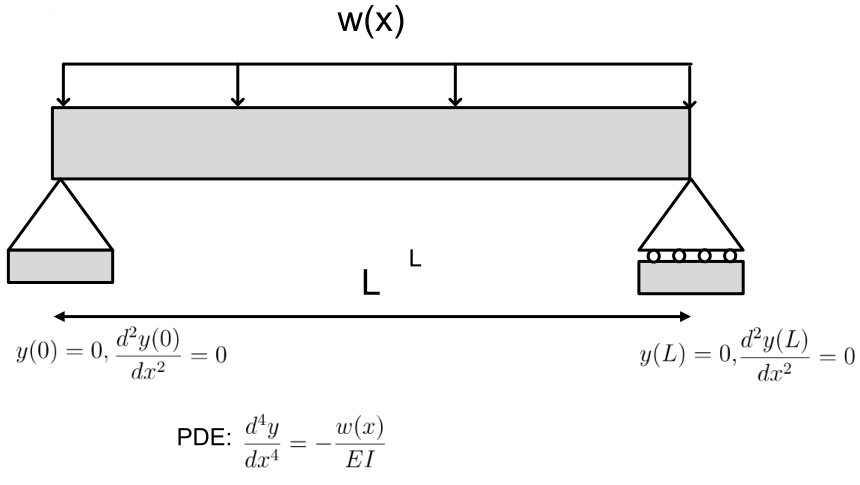

Figure 6: Schematic representations of a simply supported beam with uniformly distributed loading (Problem 2).

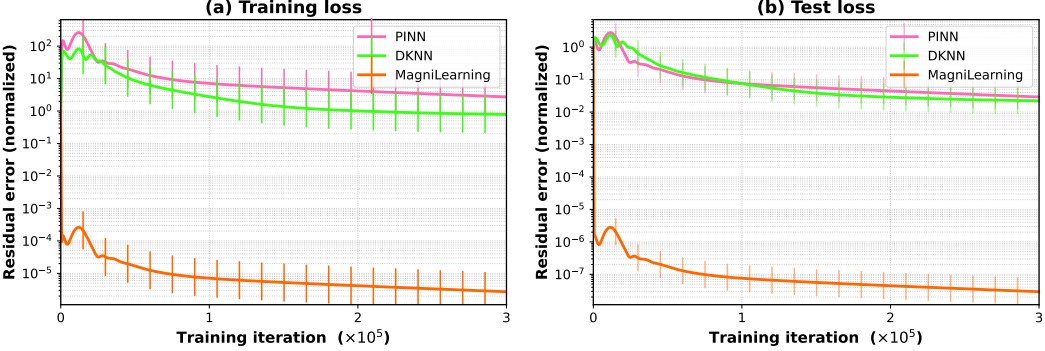

Figure 7: MagniLearning converges faster and maintains lower residuals than PINN and DKNN on Problem 2.

These boundary conditions correspond to the deflection and moment being zero at the beam's supports. The analytical solution for the deflection of the simply supported beam is:

$$y(x) = \frac{w}{24EI} \left( -x^4 + 2Lx^3 - L^3x \right).$$ (26)

The loss function for the regular PINN combines the residual loss from the governing equation and the boundary condition terms:

$$\mathcal{L}_{\text{PINN}}(x) = \text{MSE}\left( \frac{d^4y(x)}{dx^4} - \frac{w}{EI} \right) + \text{MSE}(y(0)) + \text{MSE}(y(L)) + \text{MSE}\left( \left. \frac{d^2y}{dx^2} \right|_{x=0} \right) + \text{MSE}\left( \left. \frac{d^2y}{dx^2} \right|_{x=L} \right).$$ (27)

In this problem, the bounds are chosen based on known physical properties or prior knowledge of the system:

$$u_{\text{lower}}(x) = -0.05, \quad u_{\text{upper}}(x) = 0.05.$$ (28)

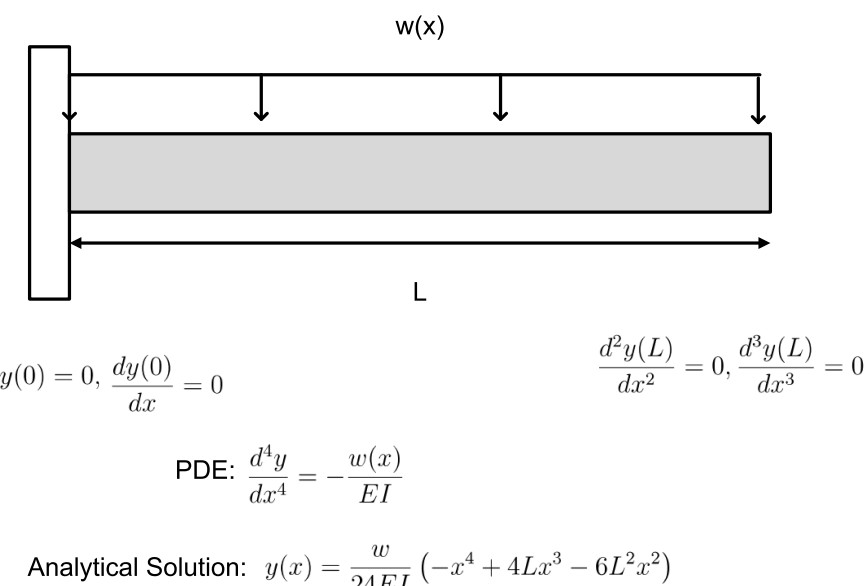

Figure 8: Schematic diagram of a Cantilever beam with uniformly distributed loading (Problem 3).

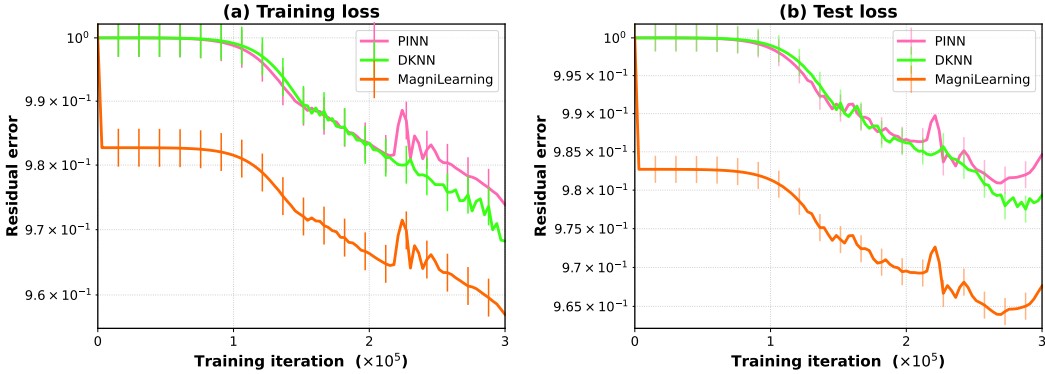

Figure 9: MagniLearning consistently maintains lower training and testing objectives than PINN and DKNN, resulting in faster and more stable convergence in Problem 3.

The domain knowledge loss penalizes predictions that exceed these bounds:

$$\mathcal{L}_{\text{DK}}(x) = \text{MSE}\left(\frac{d^4y(x)}{dx^4} - f(x)\right) + \text{MSE}(y(0)) + \text{MSE}(y(L)) + \text{MSE}\left(\left.\frac{d^2y}{dx^2}\right|_{x=0}\right) +$$

$$\text{MSE}\left(\left.\frac{d^2y}{dx^2}\right|_{x=L}\right) + \text{MSE}\left(y(x) - \text{Clamp}(y(x), u_{\text{lower}}, u_{\text{upper}})\right) \quad (29)$$

Building on this foundation, we apply MagniLearning to the simply supported beam problem, and as shown in Figure 7, the adaptive reweighting mechanism accelerates convergence and achieves lower residual errors than both PINN and DKNN, consistently leading to more stable optimization and improved predictive accuracy.

## B.3 PROBLEM 3: CANTILEVER BEAM WITH UNIFORMLY DISTRIBUTED LOAD

For a cantilever beam, the deflection can be described by the governing equations of Eqs. 22, 23, 24. For a cantilever beam of length $L$ with uniform loading $w(x) = w$, the boundary conditions are:

$$y(0) = 0, \quad \left.\frac{dy}{dx}\right|_{x=0} = 0, \frac{d^2y}{dx^2}\bigg|_{x=L} = 0, \quad \frac{d^3y}{dx^3}\bigg|_{x=L} = 0. \tag{30}$$

These boundary conditions reflect the fact that at the fixed end of the beam ($x = 0$), the deflection and slope are zero, while at the free end ($x = L$), both the moment and the shear force are zero.

The analytical solution for the deflection of the cantilever beam under uniformly distributed loading is given by Eq.26. The *loss function for the regular PINN* combines the residual loss from the governing equation and the terms of the boundary condition:

$$\mathcal{L}_{\text{PINN}}(x) = \text{MSE}\left(\frac{d^4y(x)}{dx^4} - f(x)\right) + \text{MSE}(y(0)) + \text{MSE}\left(\frac{dy}{dx}\bigg|_{x=0}\right)$$
$$+ \text{MSE}\left(\frac{d^2y}{dx^2}\bigg|_{x=L}\right) + \text{MSE}\left(\frac{d^3y}{dx^3}\bigg|_{x=L}\right). \tag{31}$$

The DKNN incorporates domain knowledge into the training process by introducing upper and lower bounds for the deflection, like in Eq.28.

The domain knowledge loss penalizes the predictions that exceed these bounds:

$$\mathcal{L}_{\text{DK}}(x) = \text{MSE}\left(\frac{d^4y(x)}{dx^4} - f(x)\right) + \text{MSE}(y(0)) + \text{MSE}\left(\frac{dy}{dx}\bigg|_{x=0}\right) + \text{MSE}\left(\frac{d^2y}{dx^2}\bigg|_{x=L}\right)$$
$$+ \text{MSE}\left(\frac{d^3y}{dx^3}\bigg|_{x=L}\right) + \text{MSE}\left(y(x) - \text{clamp}(y(x), u_{\text{lower}}, u_{\text{upper}})\right). \tag{32}$$

While these methods approximate beam responses reasonably well, they often encounter convergence inefficiencies and accuracy issues near supports or regions of high curvature. Building on this foundation, we apply MagniLearning to the simply supported beam problem, and as shown in Figure 9, its adaptive reweighting mechanism accelerates convergence and achieves lower residual errors than both PINN and DKNN, consistently delivering more stable optimization and improved predictive accuracy.

## B.4 PROBLEM 4: PROPPED CANTILEVER BEAM WITH UNIFORMLY DISTRIBUTED LOADING

The governing equation for the propped cantilever beam is the same fourth-order differential equation used in previous problems. However, the boundary conditions for the propped cantilever beam differ from those in the previous problems:

$$y(0) = 0, \quad \left.\frac{dy}{dx}\right|_{x=0} = 0, \quad y(L) = 0, \quad \frac{d^2y}{dx^2}\bigg|_{x=L} = 0. \tag{33}$$

The exact solution for the deflection of the propped cantilever beam under a uniformly distributed load is the following:

$$y(x) = \frac{w}{24EI}\left(-x^4 + \frac{5}{2}Lx^3 - \frac{3}{2}L^2x^2\right). \tag{34}$$

The loss functions for the regular PINN and the DKNN follow the same structure as discussed in the previous problems. Domain knowledge is incorporated by adding upper and lower bounds for the deflection as in Eq.11.

Building on the same fourth-order formulation, we extend MagniLearning to the propped cantilever case by adaptively reweighting loss terms across both spatial regions and training epochs. This strategy directs greater attention to the fixed and propped supports, where curvature and reaction

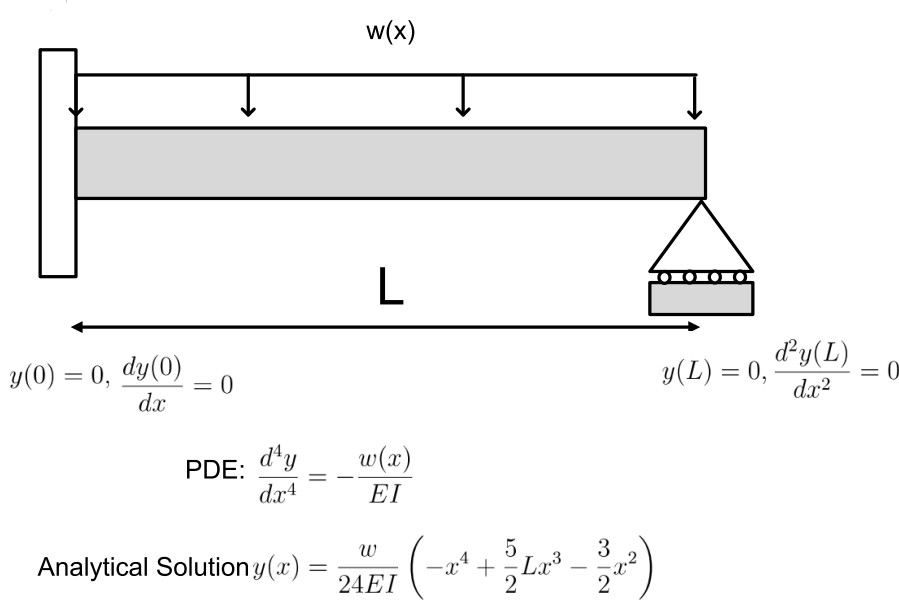

Figure 10: Schematic representations of a propped cantilever beam with uniformly distributed loading (Problem 4).

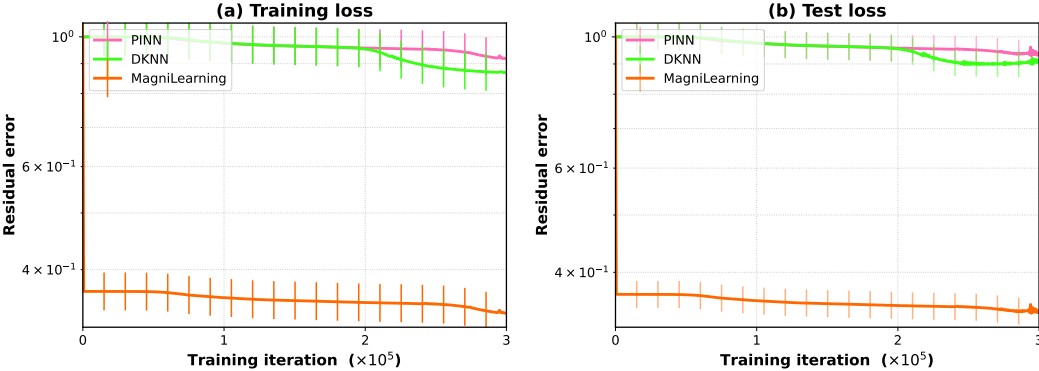

Figure 11: MagniLearning outperforms PINN and DKNN, converging faster and reaching lower error in Problem 4.

forces are most critical, while avoiding overemphasis on the less challenging mid-span regions. As illustrated in Figure 11, MagniLearning achieves faster convergence and consistently lower residual errors than both PINN and DKNN, resulting in more stable training and more accurate deflection predictions under uniformly distributed loading.

