# OpenReview forum: "Adaptive Spatial-Temporal Generalization for Physics-Informed Neural PDE Solvers"
_ICLR.cc/2026/Conference — Submitted to ICLR 2026_

### Official Review · Reviewer_ECTK · 2025-10-27

**Soundness:** 2
**Presentation:** 2
**Contribution:** 2
**Rating:** 4
**Confidence:** 2

**Summary:**

This paper proposes MagniLearning, an adaptive weighting strategy for physics-informed neural PDE solvers. MagniLearning dynamically adjusts the weights assigned to spatial regions, temporal blocks, and knowledge-based constraints during training, aiming to prioritize regions and time intervals that most influence generalization.

**Strengths:**

1. **Comprehensive Framework**: The paper develops a unified strategy (MagniLearning) that adaptively reweights loss contributions from spatial regions, temporal segments, and knowledge/incomplete supervision, using structured leave-one-out risk estimators. This offers a principled, theoretically justified way to tackle long-standing challenges in neural PDE solvers—namely, poor generalization in underrepresented or hard-to-fit regions.
2. **Theoretical Guarantees**: The work provides population risk bounds and convergence analyses (see Theorems 3.1 and 3.2, and proofs in Appendix A) that extend prior generalization results for PINNs and knowledge-augmented neural networks. These guarantees are stated with explicit dependencies on sample complexity, network width, and adaptive weighting parameters, which is valuable for researchers seeking theoretical insights into sample efficiency and over-parameterization regimes.
3. **Detailed Mathematical Derivation**: All adaptive weighting formulas and risk definitions are spelled out with clear mathematical notation (e.g., Equations for LORO/LOTO/LOKO weighting, risk bounds, and Lemma A.6). The derivations show thoughtful treatment of weighting mechanisms and their statistical effects.

**Weaknesses:**

1. **Experimental comparison and setup**: The experimental evaluation appears limited. It is essential to include comparisons with more strong and relevant baselines, such as PINNsformer [1] and RoPINN [2]. A more detailed experimental setup—following the standards set by these works—would help ensure a fair and convincing evaluation.
2. **Broader Generalization and Robustness**: How would MagniLearning perform on more complex or higher-dimensional PDEs (e.g., Navier-Stokes) and under severe noise or imperfect physics constraints? Are there scalability limitations?
3. **Ablation on Components**: Do LORO, LOKO, and LOTO contribute independently to overall performance, or is most of the gain explained by one of them? Please provide ablation or illustrative case studies per component.
4. **Sensitivity Analysis**: How sensitive is MagniLearning’s performance to the key hyperparameters ($\beta, \lambda, \kappa, \epsilon$) and normalization choices? Would performance degrade for non-optimal choices, especially in more complex or high-dimensional domains?

[1] PINNsFormer: A Transformer-Based Framework For Physics-Informed Neural Networks

[2] RoPINN: Region Optimized Physics-Informed Neural Networks

**Questions:**

See Weaknesses

---

### Official Review · Reviewer_R1G3 · 2025-10-28

**Soundness:** 1
**Presentation:** 1
**Contribution:** 1
**Rating:** 0
**Confidence:** 4

**Summary:**

This paper proposes MagniLearning, an adaptive weighting framework for physics-informed neural PDE solvers. The method integrates three strategies — Leave-One-Region-Out (LORO), Leave-One-Time-Out (LOTO), and Leave-One-Knowledge-Out (LOKO) — to dynamically adjust the importance of spatial regions, temporal segments, and knowledge components during training. The authors provide theoretical generalization bounds and demonstrate the approach on a beam mechanics problem, reporting improved convergence and stability compared to standard PINN and DKNN baselines.

**Strengths:**

The paper attempts to address the generalization and stability challenges in PINN training by introducing an adaptive reweighting mechanism for spatial and temporal domains.

The proposed framework is mathematically well-formulated, with some theoretical analysis supporting the convergence and generalization behavior.

The idea of combining spatial, temporal, and knowledge-based adaptivity into a unified framework is conceptually interesting and may inspire future exploration of dynamic weighting strategies in scientific machine learning.

**Weaknesses:**

The paper is poorly written and lacks a clear focus. While it claims to improve generalization in physics-informed neural networks (PINNs), the presentation is overly theoretical, unclear about practical contributions, and unsupported by sufficient experiments. It reads more like a mathematical exercise than a meaningful advance in the PINN literature.

1. Unclear Relationship to PINNs

Although the title and abstract emphasize “Physics-Informed Neural PDE Solvers,” most of the content is about general adaptive weighting using LORO, LOTO, and LOKO, with little explanation of how this specifically integrates with PINNs.

The methodology borrows generic ideas from “leave-one-out” validation and applies them to PDE domains, but the link to PINN loss structure is weak.

Much of the math feels copied or lightly adapted from informed learning theory (Yang & Ren, 2022) rather than addressing PDE-specific issues like boundary residuals or stiffness.

2. Overloaded and Redundant Theory

Sections 2–3 are extremely long and mathematically heavy, but most derivations are trivial restatements of existing generalization bounds (e.g., Neyshabur et al. 2015).

Many definitions (LORO, LOKO, LOTO) are repetitive and differ only in notation.

The theoretical results (Theorems 3.1, 3.2) seem symbolic rather than novel — they provide asymptotic bounds but lack meaningful intuition or experimental validation.

3. Poor Writing and Structure

The prose is verbose, repetitive, and full of grammatical errors. Sentences are long and difficult to parse; definitions are often introduced before the symbols they use are defined.

The paper’s organization jumps between probability, PDEs, and neural theory without smooth transitions.

Many references (Dong & Li 2021; Gao et al. 2025) are cited without clear explanation of how they relate to this work.

4. Extremely Limited Experiments

The “Experiments” section (Section 4) is shockingly minimal for an ICLR submission:

Only one main problem (“Uniformly Distributed Axial Load on a Rod”) is tested.

Claims of “four distinct beam problems” are not supported by figures or data — only one or two plots are shown.

The baselines (PINN, DKNN) are not strong or modern. No comparisons to well-established PINN variants (e.g., FBPINN, cPINN, XPINN) are provided.

The hyperparameter settings (β = 0.5, λ = 0.5) appear arbitrary, with no sensitivity analysis.

5. Lack of Conceptual Novelty

The proposed “MagniLearning” essentially performs region/time importance reweighting, which is similar to curriculum learning or sample reweighting — not new in the PINN context.

There is no new architecture, no new loss formulation beyond weighting existing terms, and no clear theoretical insight about PDE structure.

The adaptive weighting formula (based on exponential scaling of local errors) is heuristic, not physically grounded.

**Questions:**

See weakness

---

### Official Review · Reviewer_kFEt · 2025-10-28

**Soundness:** 2
**Presentation:** 1
**Contribution:** 1
**Rating:** 2
**Confidence:** 2

**Summary:**

The authors propose MagniLearning, an adaptive weighting framework for training physics-informed neural PDE solvers. The core idea is to dynamically adjust the importance of different loss components during training. They propose a unified framework that allow different strategies: Leave-One-Region-Out, Leave-One-Knowledge-Out and Leave-One-Time-Out.

They provide theoretical generalization bounds for the LORO and LOKO components and demonstrate experimentally on a set of 1D beam problems that MagniLearning converges significantly faster and to a lower residual error than standard PINN and DKNN baselines.

**Strengths:**

The idea of using a "leave-one-out" scheme to re-weight the loss components is interesting to allow the model to focus on challenging components.

The authors provide theoretical justifications.

**Weaknesses:**

The paper suffers from several major weaknesses.

First, the paper is extremely hard to understand due to its presentation. A lot of figures are present but they are not referenced anywhere in the text. Some sentences make no sense. This is very hard to understand the method due to the number of symbols used and how the text is presented in general. It seems that the paper has been rushed.

The experiments considered are weak and does not support the claims made. The paper claims to solve PDEs, but it has been tested on simple linear ODEs. It claims spatio-temporal generalization but it has not been tested on such cases.  The generalization experiments are very low, where only the load parameter has been varied. This does not reflect the real motivations of the paper, which refer to high-frequency, multi-scale and large-scale problems.

Computational costs of the methods have not been presented. It is important to report them to correctly assess the method.

**Questions:**

: Can the authors provide results on at least one 2D or 3D PDE problem? The 1D ODEs presented are not convincing benchmarks for a method intended for general PDE solvers.

Please provide a thorough analysis of the computational complexity and wall-clock training time of MagniLearning versus the baselines.

**Details Of Ethics Concerns:**

No ethical concerns.

---

### Official Review · Reviewer_SMLC · 2025-11-07

**Soundness:** 2
**Presentation:** 2
**Contribution:** 2
**Rating:** 2
**Confidence:** 3

**Summary:**

This paper proposes MagniLearning, an adaptive framework for improving generalization in neural PDE solvers, which integrates data-driven and knowledge-based supervision in scientific machine learning. The method dynamically adjusts the importance of different data subsets, spatial regions, temporal segments, and knowledge terms through three mechanisms: Leave-One-Region-Out (LORO), Leave-One-Time-Out (LOTO), and Leave-One-Knowledge-Out (LOKO). These adaptive weights prioritize underrepresented or informative subsets, theoretically reducing generalization error. An adaptive scheduling strategy further shifts training emphasis from data fitting to physics-based regularization over time.

**Strengths:**

1. The authors provide rigorous theoretical analysis that attempt to bound generalization risk under the proposed adaptive weighting scheme.

2. The framework explicitly integrates both label-based and knowledge-based supervision and proposes a dynamic schedule to shift emphasis from labels to knowledge during training, which is a useful design for problems where labeled data are limited but physics knowledge exists.

**Weaknesses:**

1. The experimental scope is too limited and narrow: evaluations are limited to four beam problems with relatively simple physics and low-dimensional PDEs, which weakens claims that MagniLearning generalizes broadly to complex or high-dimensional PDEs (e.g., Navier-Stokes turbulence, multiphysics, or 3D domains).

2. The baseline set is too limited. Comparing only to PINN and DKNN omits many relevant, modern baselines (e.g., domain-decomposed PINNs, XPINNs, adaptive curriculum or point-weighting schemes, operator learning methods, and other recent PINN variants). This makes it difficult to judge how much of the observed improvement is specific to MagniLearning versus attributable to baseline weaknesses.

3. The computational and practical feasibility of the core weighting mechanisms is insufficiently addressed. LORO/LOTO/LOKO as described require computing model variants with regions/times/knowledge held out (which can be costly); the manuscript lacks concrete approximations, influence-function alternatives, or runtime measurements to show how this scales in practice.

4. The claimed theoretical guarantees depend on strong, opaque technical conditions (many asymptotic/overparameterization bounds with complex dependencies). The results are difficult to interpret in realistic regimes, and the paper does not provide guidance on whether practical network sizes/ϕ choices satisfy these assumptions.

5. The presentation and exposition are often dense and stylistically choppy. Notation is heavy, several definitions are introduced with minimal intuition, and transitions between theory, algorithm, and experiments are abrupt. This makes it harder for readers to connect the theoretical claims to empirical practice.

6. The ML contribution is insufficiently emphasized relative to the application contribution. Much of the novelty lies in the choice of adaptive weighting rather than a fundamentally new learning algorithm. The manuscript needs to better articulate the general ML insights that would interest the ICLR community.

7. Key ablations are missing. The paper does not isolate the relative contribution of LORO vs LOTO vs LOKO, nor does it quantify sensitivity to weight-scaling hyperparameters (β, λ, τ, κ) or to the ε-net region radius ϕ, which limits understanding of robustness and reproducibility.

8. Training and implementation details are sparse and seem lightweight for a strong empirical claim: the experiments use a very small MLP (two layers × 50 units) and a single optimizer setting (Adam, lr=1e-2, 2000 epochs) without reporting hyperparameter sweeps, variance across seeds, training time, or cost of adaptive weighting. This undercuts claims about scalability and reliability.

9. The manuscript claims broad applicability and improved generalization but does not demonstrate transfer experiments (e.g., training on one geometry/parameter regime and testing on another) nor does it show performance on noisy or sparse label regimes in a way that convincingly separates the benefit of adaptive weighting from simple reweighting heuristics.

**Questions:**

1. How frequently is the adaptive graph recomputed during training and inference, and how does this affect computational cost and stability?

2. Can the authors provide theoretical or empirical evidence that the adaptive graph construction leads to better operator generalization rather than simply overfitting to localized regions?

3. How does the proposed temporal attention mechanism compare to existing time-stepping adaptive schemes, such as variable Δt integrators or latent ODE-based operator models?

4. Would the adaptive mechanism remain effective when applied to 3D turbulent flows or other complex systems with significantly higher resolution requirements?

5. How sensitive is the performance to the hyperparameters controlling graph update frequency, attention temperature, or neighbor sampling?

---

### Meta-Review · Area_Chair_yi6R · 2025-12-30

**Summary:**

This work aims to present magnilearning, an adaptive reweighing mechanism for physics-informed machine learning. The reviewers have criticized several aspects of the paper -- the framework, the presentation, the very thin experimental base of ODEs and the lack of comparison to modern PINN methods. As the authors did not defend their work by engaging in a discussion, the paper cannot be accepted in current form.

**Reviewer Concerns:**

Not applicable as the authors did not participate in the discussion.

**Reviewer Scores:**

Not applicable as the authors did not participate in the discussion.

---

### Decision · Program_Chairs · 2026-01-26

Reject